# Polysome-CAGE of TCL1-driven chronic lymphocytic leukemia revealed multiple N-terminally altered epigenetic regulators and a translation stress signature

Ariel Ogran[1], Tal Havkin-Solomon[1], Shirly Becker-Herman[2], Keren David[2], Idit Shachar[2], Rivka Dikstein[1]*

[1]Department of Biomolecular Sciences, The Weizmann Institute of Science, Rehovot, Israel; [2]Department of Immunology, The Weizmann Institute of Science, Rehovot, Israel

*For correspondence:
rivka.dikstein@weizmann.ac.il

Competing interest: The authors declare that no competing interests exist.

**Abstract** The transformation of normal to malignant cells is accompanied by substantial changes in gene expression programs through diverse mechanisms. Here, we examined the changes in the landscape of transcription start sites and alternative promoter (AP) usage and their impact on the translatome in TCL1-driven chronic lymphocytic leukemia (CLL). Our findings revealed a marked elevation of APs in CLL B cells from Eµ-Tcl1 transgenic mice, which are particularly enriched with intra-genic promoters that generate N-terminally truncated or modified proteins. Intra-genic promoter activation is mediated by (1) loss of function of 'closed chromatin' epigenetic regulators due to the generation of inactive N-terminally modified isoforms or reduced expression; (2) upregulation of transcription factors, including *c-Myc*, targeting the intra-genic promoters and their associated enhancers. Exogenous expression of *Tcl1* in MEFs is sufficient to induce intra-genic promoters of epigenetic regulators and promote *c-Myc* expression. We further found a dramatic translation downregulation of transcripts bearing CNY cap-proximal trinucleotides, reminiscent of cells undergoing metabolic stress. These findings uncovered the role of *Tcl1* oncogenic function in altering promoter usage and mRNA translation in leukemogenesis.

## Editor's evaluation

Results presented in this article suggest that T-Cell Leukemia/Lymphoma 1 (TCL1) protein promotes alternative transcription site selection in chronic lymphoid leukemia, resulting in the production of N-terminally truncated versions of chromatin regulators and induction of expression of transcription factors including c-MYC. Finally, the authors provide evidence that TCL1 drives translational reprogramming akin to that observed under metabolic stress. Notwithstanding technical limitations that obscured direct detection of N-terminally truncated protein variants, it was found that this study is of broad potential interest inasmuch as it provides previously unappreciated evidence that TCL1 orchestrates epigenetic, transcriptional, and translational rewiring in leukemic cells. Based on this, the study by Ogran et al. should be of significant interest across a number of biomedical research disciplines ranging from regulation of gene expression to cancer research.

## Introduction

Chronic lymphocytic leukemia (CLL) accounts for 25–30% of all leukemias in the Western countries, with incidence rates ranging from 3.65 to 6.75 cases per 100,000 population per year (*Siegel*

*et al., 2019*; *Sant et al., 2010*). CLL is characterized by an outgrowth of malignant CD5 positive B cells, mainly residing in the peripheral blood (PB), bone marrow, and lymphoid organs, and by a high biological heterogeneity reflected in clinically different outcomes including disease progression, therapy response, and relapse (*Kröber et al., 2002*; *Zenz et al., 2010*). The hallmark of the CLL disease is mainly decreased apoptosis resulting from overexpression of anti-apoptotic proteins and resistance to apoptosis in vivo. The *Tcl1* oncogene is highly expressed in CLL cells, and its expression level correlates with the CLL aggressiveness (*Herling et al., 2006*). *Tcl1* was initially discovered as a gene involved in the rearrangement of the T-cell leukemia/lymphoma 1 locus at 14q32.1, the most common chromosomal aberrations detected in mature T-cell leukemias (*Virgilio et al., 1994*). The central role of TCL1 in CLL was demonstrated in transgenic mice overexpressing the human *Tcl1* gene under the control of the immunoglobulin promoter and enhancer (Eμ-Tcl1) (*Bichi et al., 2002*). These mice develop normally into adulthood but develop CLL that is highly similar to the human disease and is characterized by CD5+ cells, which accumulate in the spleens, livers, and lymph nodes.

The association of high TCL1 expression and malignant transformation is linked to its involvement in several major oncogenic pathways, including the enhancement of AKT activity (*Herling et al., 2008*; *Noh et al., 2012*; *Patil et al., 2020*; *Pekarsky et al., 2010*); interaction with the DNA repair ataxia–telangiectasia-mutated (ATM) factor (*Gaudio et al., 2012*), and the transcription factor CREB (*Pekarsky et al., 2008*) as well as in contributing to accelerated tumorigenic and the prosurvival NF-κB signaling (*Gaudio et al., 2012*). Inhibition of activator protein 1 (AP-1) transcriptional activity via interaction of TCL1 with the AP-1 complex represents another mechanism to antagonize the expression of pro-apoptotic factors (*Gaudio et al., 2012*; *Motiwala et al., 2011*). TCL1 was also shown to contribute to epigenetic reprogramming via interacting with the de novo DNA methyltransferase 3A (DNMT3A) and reducing its enzymatic activity (*Palamarchuk et al., 2012*). Considering the central role of TCL1 in tumor initiation, progression, and maintenance, it is reasonable that its oncogenic functions involve additional, yet to be discovered, activities.

The transcription initiation of a specific gene can occur from more than one position in the DNA, creating several mRNA isoforms. Alterations in transcription start site (TSS) selection and alternative promoter (AP) usage increase transcriptome diversity and regulation. For example, the level of transcription initiation can vary between different TSSs under different growth conditions, in response to a specific signal, or different cell types and tissues. In addition, mRNA isoforms with different 5′ leaders can vary in their translation efficiency (TE) or half-lives. Likewise, AP usage can lead to the generation of protein isoforms that differ in their N termini and as a result can have different or even opposite biological functions. Recent large-scale promoter analysis in hundreds of human and mouse primary cell types shed light on the prevalence of AP usage in mammals (*Forrest et al., 2014*) as well as in multiple cancer types (*Demircioğlu et al., 2019*). Several studies have examined the translation and stability of transcript isoforms of the same gene and the contribution of AP usage to the translational response to stress (*Floor and Doudna, 2016*; *Arribere and Gilbert, 2013*; *Wang et al., 2016*; *Tamarkin-Ben-Harush et al., 2017*). However, presently little is known about the impact of TSSs and AP usage changes on the protein content of cells undergoing cancer progression. Considering that even transient cellular stress is associated with dramatic changes in the landscape of TSSs and AP usage that are coordinated with translation (*Tamarkin-Ben-Harush et al., 2017*), it is reasonable to assume that such alterations can affect translatability in cells undergoing oncogenic transformation. Some of the new cancer-specific protein isoforms generated by AP usage can help maintain the transformed phenotype. Presently, the vast majority of studies measured global changes in mRNA abundance and transcript translatability in cancer cells, while the potential effect of variations in TSSs selection and AP usage on translation was hardly addressed.

This study examined the changes in TSS selection and AP usage and their impact on the translatome during malignant transformation of B cells by the *Tcl1* oncogene. Using healthy and Eμ-Tcl1 transgenic mice, we compared the TSSs landscape of normal and CLL B cells. Our findings revealed a marked elevation of APs in malignant CLL B cells with a particularly high prevalence of intra-genic promoters that are predicted to generate N-terminally truncated or N-terminally modified proteins. The induced cryptic promoters are driven by nearby enhancers and upregulation of their cognate transcription factors, including the *c-Myc* oncogene. A promoter shift in the opposite direction accounts for the induced expression of the immune checkpoint ligand PD-L2. Notably most of the intra-genic-generated transcripts are also efficiently translated. These remarkable changes of promoters and TSSs

are linked to the loss of function of many 'closed chromatin' epigenetic regulators via induction of inactive isoforms by promoter shifts or by reduced expression, resulting in a feed-forward loop. The activation of intra-genic cryptic promoters and the *c-Myc* oncogene elevation are mediated, at least in part, by the *Tcl1* oncogene itself and are therefore intrinsic to the CLL B cells. We further found a dramatic and specific translation downregulation of transcripts bearing CNY cap-proximal trinucleotides, reminiscent of cells undergoing energy stress. These findings explored the contribution of 5′ end mRNA isoforms and their associated translational changes during cellular transformation and uncovered novel *Tcl1* oncogenic pathways.

## Results

### Global analysis of TSSs revealed extensive induction of promoter shifts in CLL B cells

To address the potential contribution of AP to CLL we used the Eμ-Tcl1 transgenic mice overexpressing the human *Tcl1* gene under the control of the immunoglobulin heavy chain (IgH) variable region promoter (V$_H$) and immunoglobulin heavy chain enhancer (Eμ-Tcl1) (*Bichi et al., 2002*). Like the human CLL, Eμ-Tcl1 mice accumulate transformed CD5+ B cells in the spleens, livers, and lymph nodes during adulthood, reproducing CLL (*Pekarsky et al., 2008*). Splenocytes from healthy and Eμ-Tcl1 mice were subjected to affinity-based capture using CD19 magnetic beads to isolate B cells and CLL B cells, respectively. Total RNA was extracted and subjected to Cap Analysis of Gene Expression (CAGE) (*Carninci et al., 2006*; *Takahashi et al., 2012*) library preparation and Illumina deep sequencing of the first 27 nucleotides CAGE tags from the 5′ ends (*Figure 1A*). An average of 10.625 million CAGE tags per sample was aligned to a total of 2.67 million CAGE-derived TSSs (CTSSs) that were mapped and clustered to a consensus set of 28,616 promoters that correspond to a total of 11,515 genes. Using this high-throughput data from two independent biological replicates, we determined the global landscape of TSSs and their relative abundance in healthy and CLL B cells. Meta-gene analysis of both WT and Eμ-Tcl1 CAGE data showed almost perfect alignment to the annotated TSS atlas of the FANTOM5 project (*Lizio et al., 2015*; *Figure 1B*), indicating the proper calculation of the TSS positions at a single-base resolution. The CAGE libraries were assessed with coefficient scores calculated using the Pearson pairwise-correlation test and the reproducibility of the replicates were found to be high (*Figure 1—figure supplement 1A*). As expected, in both WT and Eμ-Tcl1 samples, most of the TSSs with the highest expression level were mapped to the core promoter region (*Figure 1C*). A significant fraction of the TSSs was found within introns and intergenic regions. Since the CAGE method captures all capped RNAs, these data also detect enhancer RNAs (eRNAs), a class of short noncoding RNAs resulting from a balanced bidirectional transcription from enhancers that are unstable and nonpolyadenylated (*Andersson et al., 2014*; *Kim et al., 2010*) (examples of an enhancer and super-enhancer are shown in *Figure 1—figure supplement 1A, C*). We examined the possibility that part of the intronic and intra-genic CAGE reads are derived from active eRNAs. Based on the criteria of balanced bidirectional read mapping, we calculated the frequencies of eRNAs in introns and intergenic regions and found that most of the intronic TSSs (85% in Eμ-Tcl1 and 79% in WT) and a significant fraction of the intra-genic TSSs (62% in Eμ-Tcl1 and 50% in WT) are indeed derived from eRNAs (*Figure 1C*). The ability of CAGE to detect lowly expressed genes makes it crucial to reduce the mean–variance relationship of count data prior to differential expression (DE) analysis. A 'blind' analysis of the variance stabilizing transformation was performed and summarized by a principle component analysis (PCA). PCA1 explained 85% of the variance and distinguished between WT and Eμ-Tcl1, where WT samples are tightly clustered and Eμ-Tcl1 are separated by 13% of the variance, which is explained by PCA2 (*Figure 1—figure supplement 1D*). As no systemic effect on samples was found, we decided to fit a DE model without using a correction factor. Out of the 28,616 consensus TSSs, we found in Eμ-Tcl1 3337 upregulated and 1829 downregulated TSSs (*Figure 1D*), suggesting an overall upregulation of many promoters in CLL. To test the contribution of eRNA to the differentially expressed TSSs in Eμ-Tcl1, we used an established method to predict eRNA–TSS interactions based on genomic distance and expression-based correlation (*Andersson and Sandelin, 2020*; *Thodberg and Sandelin, 2019*). We found 255 upregulated eRNAs, of them 106 were positively correlated to the 203 upregulated TSSs, while out of 179 downregulated eRNAs, 86 were positively correlated to the 331 downregulated TSSs (*Figure 1D*). Lengths of the TSS clusters were calculated by

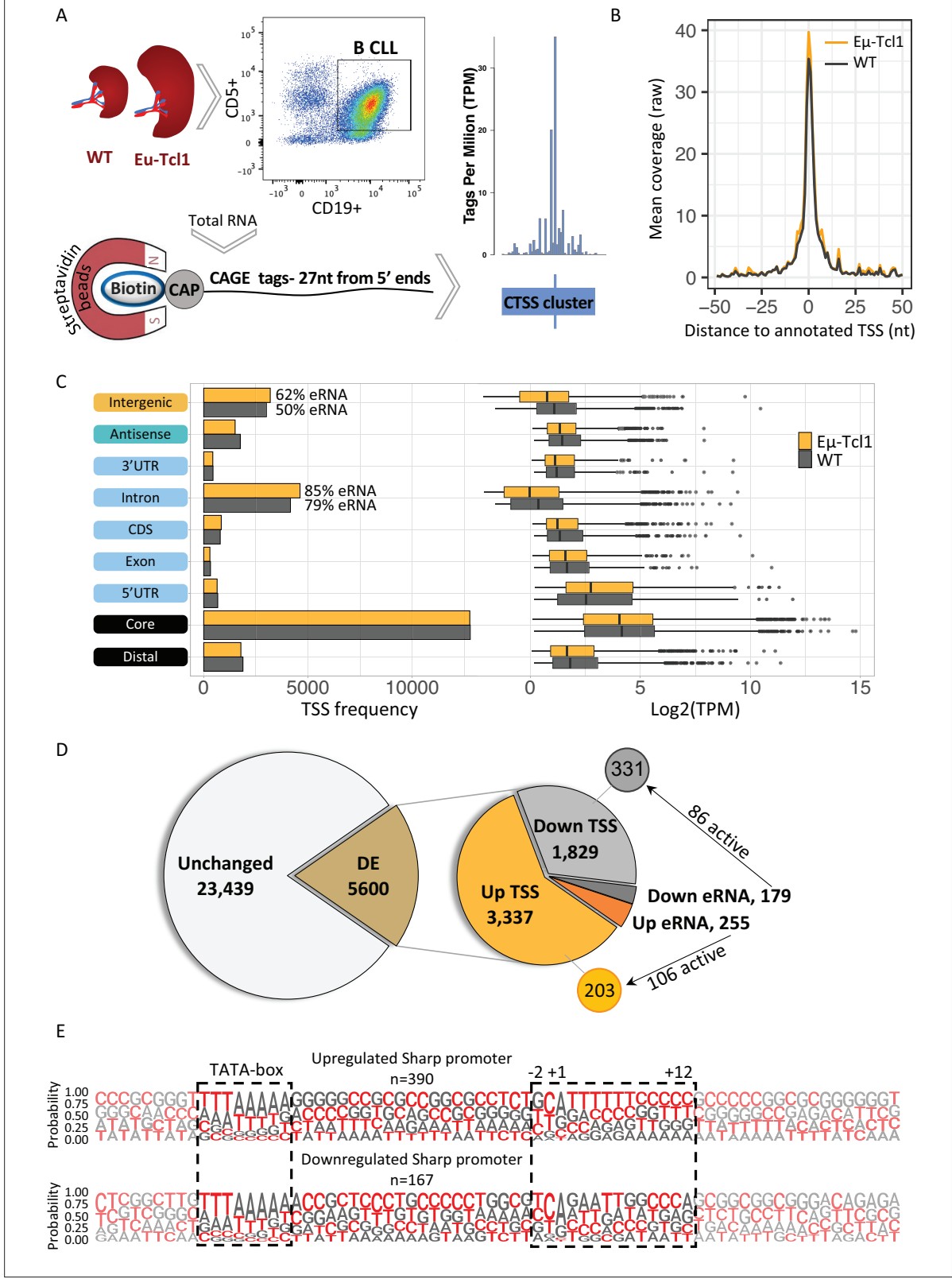

**Figure 1.** Global analysis of transcription start sites (TSSs) in healthy and chronic lymphocytic leukemia (CLL) B cells. (**A**) A scheme of the Cap Analysis Gene Expression (CAGE) in healthy (WT) and CLL B cells (Eµ-Tcl1). WT and Eµ-Tcl1 splenic B cells were isolated using CD5 and CD19 magnetic beads. Total RNA samples were subjected to the CAGE as described in Materials and methods. CAGE-derived TSSs (CTSSs) were mapped and clustered into genomic blocks referred as tag clusters (blue bar) with a peak of most recovered CTSS (blue tick) indicates the TSSs in a single-base resolution. (**B**)

*Figure 1 continued on next page*

*Figure 1 continued*

Metagene analysis of Eµ-Tcl1 (orange) and WT (gray) CAGE libraries aligned to annotated TSS atlas of the FANTOM5 project. (**C**) The frequency (left) and expression level (right, log2 TPM) of CAGE tag clusters in Eµ-Tcl1 (orange) and WT (gray) mice by gene locations. The percentage values on the columns of intronic and intergenic locations refer to predicted enhancer RNA (eRNA) TSSs. (**D**) Differentially expressed (DE) TSSs. Left circle indicates unchanged (light gray) and significantly DE (brown) TSS between Eµ-Tcl1 and WT mice. Right circle details frequencies of Eµ-Tcl1 upregulated genic and eRNA TSSs (orange and dark orange, respectively) and downregulated genic and eRNA TSSs (gray and dark gray). Subgrouping of DE TSSs that are positively correlated with enhancers are indicated. (**E**) Sequence LOGOs (−40 to +30 relative to the TSS) of Eµ-Tcl1 upregulated (lower panel) and downregulated (upper panel) sharp promoters where TATA box situated between −31 and −24 sites and cap-proximal region (12 bases) are boxed in a dashed line.

The online version of this article includes the following figure supplement(s) for figure 1:

**Figure supplement 1.** Global analysis of transcription start sites (TSSs) in healthy and chronic lymphocytic leukemia (CLL) B cells.

the interquartile range (IQR) of highly expressed (>10 tag per million [TPM]) TSSs. TSS clusters lengths holding 10–90% of pooled TSSs expression were found to distribute into sharp and broad promoters bimodally (*Figure 1—figure supplement 1E*). Consistent with known promoter architecture (*Carninci et al., 2006*; *Suzuki et al., 2001*), general TSSs are characterized by YR at the (−1) and (+1) positions (*Figure 1—figure supplement 1E*). Promoters with broad TSS distributions tend to be surrounded by CpG islands, while promoters with sharp TSSs are defined by the presence of a TATA box around position −30 (*Figure 1—figure supplement 1F*). Separately plotted LOGOs of differentially expressed TSSs, showed enrichment of poly-pyrimidine sequence in the upregulated sharp TSSs at the positions of +2 to +12 (*Figure 1E*).

## A marked elevation of new CLL-specific protein isoforms generated by intra-genic TSSs

When an alternative TSS is located at the promoter or at the 5′ UTR regions, the original ORF is retained. However, new intra-genic TSSs located in introns or CDS are predicted to give rise to protein isoforms bearing alternative or truncated N termini and, consequently, potential different or even opposite biological functions (illustrated scheme in *Figure 2A*). We analyzed AP usage in our data and found that 2530 genes (22%) contain more than one promoter that contributes over 10% of their total gene expression (*Figure 2B*), consistent with previous studies (*Tamarkin-Ben-Harush et al., 2017*). To determine differential TSS usage (DTU), we examined the multipromoter subset and found 550 significant DTUs from 489 genes where the alternative TSS is upregulated in Eµ-Tcl1 and 373 DTUs from 328 genes with downregulated alternative TSS. The 47% more cases of upregulated alternative TSS indicate that DTU during CLL transformation is gaining more new isoforms than losing the canonical ones (*Figure 2C*). The Eµ-Tcl1 upregulated alternative TSSs were enriched at the core promoter, 5′ UTR, CDS, and intron regions but not at distal regions (*Figure 2C*). Median expression (log2 TPM) of alternative TSS in all locations ranged between 1.6 and 2.7 with no apparent differences between up- and downregulated alternative TSS (*Figure 2—figure supplement 1A*). By analyzing the correlation between alternative TSSs and nearby active enhancers, which are marked by eRNAs, a clear elevation in eRNA frequencies was found in Eµ-Tcl1 upregulated alternative TSSs (*Figure 2—figure supplement 1B*). These findings suggest the significant role of active enhancers in the activation of DTUs in Eµ-Tcl1.

To explore the consequences of DTUs in CLL, we first analyzed several DTU events (*Figure 2D–F*). An intriguing example of DTU resulting in an N-terminally truncated isoform was found in programmed cell death 1 ligand 2 (*Pdcd1lg2*/PD-L2). This gene has an intra-genic alternative TSS generating a protein isoform lacking the Ig-like extracellular domains of PD-L2 (*Latchman et al., 2001*; *Li et al., 2018*; *Zak et al., 2017*), specifically expressed in WT B cells and is predicted to be inactive. In contrast, the full-length protein is exclusively expressed in the CLL cells via promoter shift and likely contributes to immunosuppression (*Figure 2D*). An opposite example is chromodomain 1 (*Chd1*), in which promoter shifting in Eµ-Tcl1 is within an intra-genic alternative TSS that results in truncation of several critical active domains, presumably leading to loss of function (*Figure 2E*). The F-box only protein 5 (*Fbxo5*) has two promoters located upstream to the main ORF differing only by their 5′ UTR length resulting in 490 nt shorter 5′ UTR in the Eµ-Tcl1-specific transcript isoform (*Figure 2F*). Global analysis of the promoter shifts levels and its significance of all DTUs revealed that the upregulated alternative TSS isoforms predicted to give rise to proteins with alternative/truncated N termini are dramatically

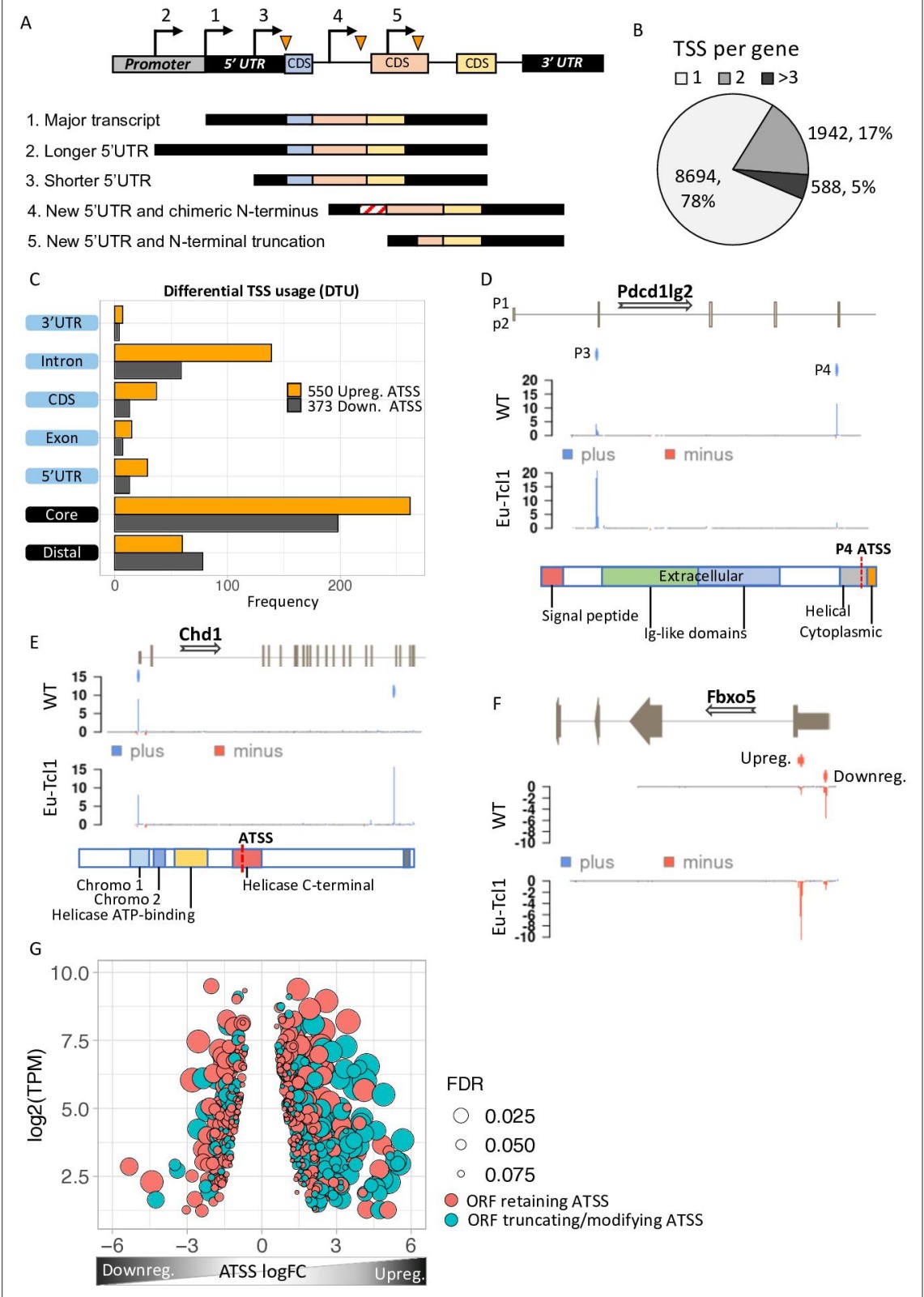

**Figure 2.** Analysis of differential TSS usage (DTU) reveals alternative 5′ UTRs and new protein isoforms derived from cryptic promoters in chronic lymphocytic leukemia (CLL). (**A**) Scheme of five alternative TSSs (ATSSs) generating isoforms with alternative 5′ UTR regions (*Siegel et al., 2019*; *Sant et al., 2010*; *Kröber et al., 2002*; *Zenz et al., 2010*; *Herling et al., 2006*) and modification of the N′ termini for example, new chimeric CDS (*Kröber et al., 2002*) and N′-terminal truncation (*Herling et al., 2006*). (**B**) The frequency and the percentage of uni-TSS genes and multi-TSS genes

*Figure 2 continued on next page*

Figure 2 continued

including only TSS clusters that constitute over 10% of total gene expression. (**C**) Frequencies of alternative TSSs that were upregulated (orange) and downregulated (gray) in Eµ-Tcl1, grouped by gene-structure locations. (**D**) Gene track showing differential TSS usage (DTU) in *Pdcd1lg2* (PD-L2) in Eµ-Tcl1 and WT mice. Canonical promoter (P3) and intra-genic promoter (P4, located just upstrem to the last CDS of PD-L2) are up- and downregulated in Eµ-Tcl1, respectively. Both, located on the plus strand (blue). Bellow, is a scheme illustrating the P4-induced truncation of PD-L2 lossing most of its functional domains. (**E**) *Chd1* protein truncation in Eµ-Tcl1 resulted by induction of cryptic promoter. In the upper track, two promoters on the plus strand (blue ticks) presented along to a partial gene-structure scheme. While the canonical promoter located upstream to the annotated 5' UTR is similarly expressed in Eµ-Tcl1 and WT, an Eµ-Tcl1-specific induction of an intra-genic alternative TSS located in the 16th intron resulting in N' terminus truncation of Chd1 lacking several active domains as illustrated below. (**F**) *Fbxo5* 5' UTR shortening resulted by DTU in Eµ-Tcl1. On the upper track, two alternative promoters located on the minus strand shown as short red bars with indicator tick of TSS. Coverage peaks in the two lower tracks (WT and Eµ-Tcl1) showing differential expression of the two promoters where the nearest alternative TSS to the ORF is upregulated in Eµ-Tcl1. (**G**) Dot plot of ORF-retaining alternative TSS (ATSS) in peach and ORF-truncating intra-genic alternative TSS (turquoise) presenting expression (log2 TPM) against ATSS log FC evaluating the degree of promoter shifting (DTU). Positive and negative ATSS log FCs refer to Eµ-Tcl1 up- and downregulated ATSS, respectively. Dot sizes correspond to the FDR statistical significance of the DTU analysis.

The online version of this article includes the following figure supplement(s) for figure 2:

**Figure supplement 1.** Analysis of differential TSS usage (DTU) reveals alternative 5'' UTRs and new protein isoforms derived from cryptic promoters in chronic lymphocytic leukemia (CLL).

increased in the Eµ-Tcl1 samples compared to WT B cells (*Figure 2G*). This finding suggests a previously unappreciated diversity of the CLL proteome.

Gene ontology (GO) analysis of genes affected by AP usage shows enrichment in critical CLL pathways, for example, NF-κB signaling (*Pekarsky et al., 2008*), MAPK signaling (*Shukla et al., 2018*), Toll-like receptor signaling (*Spaner and Masellis, 2007*), phosphatidylinositol 3-kinase signaling (*Barragán et al., 2002*), chromatin regulation/acetylation, and other relevant GOs (*Supplementary file 1*). *Supplementary file 2* lists the phenotypes derived from mouse genome informatics (MGI) associated with the DTU gene subset, such as increased B-cell number, enlarged spleen, and thymus hypoplasia, consistent with hematopoietic malignancy.

## Tcl1 promotes chromatin relaxation, induction of intra-genic cryptic promoters, and activation of the *c-Myc* oncogene

To elucidate the underlying basis for the broad induction of intra-genic cryptic promoters, we considered previous genetic and molecular studies reporting transcription initiation within coding regions as a consequence of defects in the activities of chromatin regulators that maintain closed chromatin structure or nucleosome positioning (*Brocks et al., 2017*; *Wei et al., 2019*; *Mozzetta et al., 2015*; *Hennig and Fischer, 2013*; *Wei et al., 2020*; *Hennig et al., 2012*). Our Eµ-Tcl1 data found 34 chromatin remodelers with ORF-retaining or -truncating alternative TSSs that could be classified into factors associated with open or close chromatin states, 28 of them were upregulated in Eµ-Tcl1 (*Figure 3A*). Remarkably, most of the upregulated alternative TSS within coding regions of chromatin modifiers, such as the known key epigenetic gatekeepers such as *Hdac4*, *Hdac5*, *Sirt2*, *Smyd4*, *Kdm4a*, *Dnmt3a*, *Chd1*, and *Mgmt*, are predicted to express truncated proteins that cause loss of critical domains due to DTU (*Figure 3A* and 5D). To validate the CAGE data, we picked four chromatin modifiers and analyzed them by 5' RACE-qPCR using RNA samples from Eµ-Tcl1 mice CLL B cells. The results of the 5'RACE show dramatic upregulation of the isoforms originating by alternative TSS of the four selected Dnmt3a, *Chd1*, *Kdm4a*, and *Sirt2* (*Figure 3—figure supplement 1*). In addition to the truncated chromatin modifiers, numerous other epigenetic regulators of closed chromatin structure are strongly downregulated (*Table 1*). These findings revealed widespread chromatin deregulation in Eµ-Tcl1 B cells that can activate cryptic promoters.

The induction of intra-genic promoters can be either driven by TCL1 (intrinsic) or by environmental signals associated with cancer progression. To distinguish between these possibilities, we established three mouse embryonic fibroblasts (MEFs) clones stably expressing human TCL1, and determined by RT-qPCR the relative expression of the alternative intra-genic TSS of four selected genes (as shown in the scheme). We observed a significant elevation of the intra-genic truncating TSS of *Hdac5*, *Dnmt3a*, and *Kdm4a* but not *Zswim4*. These findings suggest that *Tcl1* itself can promote relaxation of the chromatin structure and promoter shifts, at least in some genes.

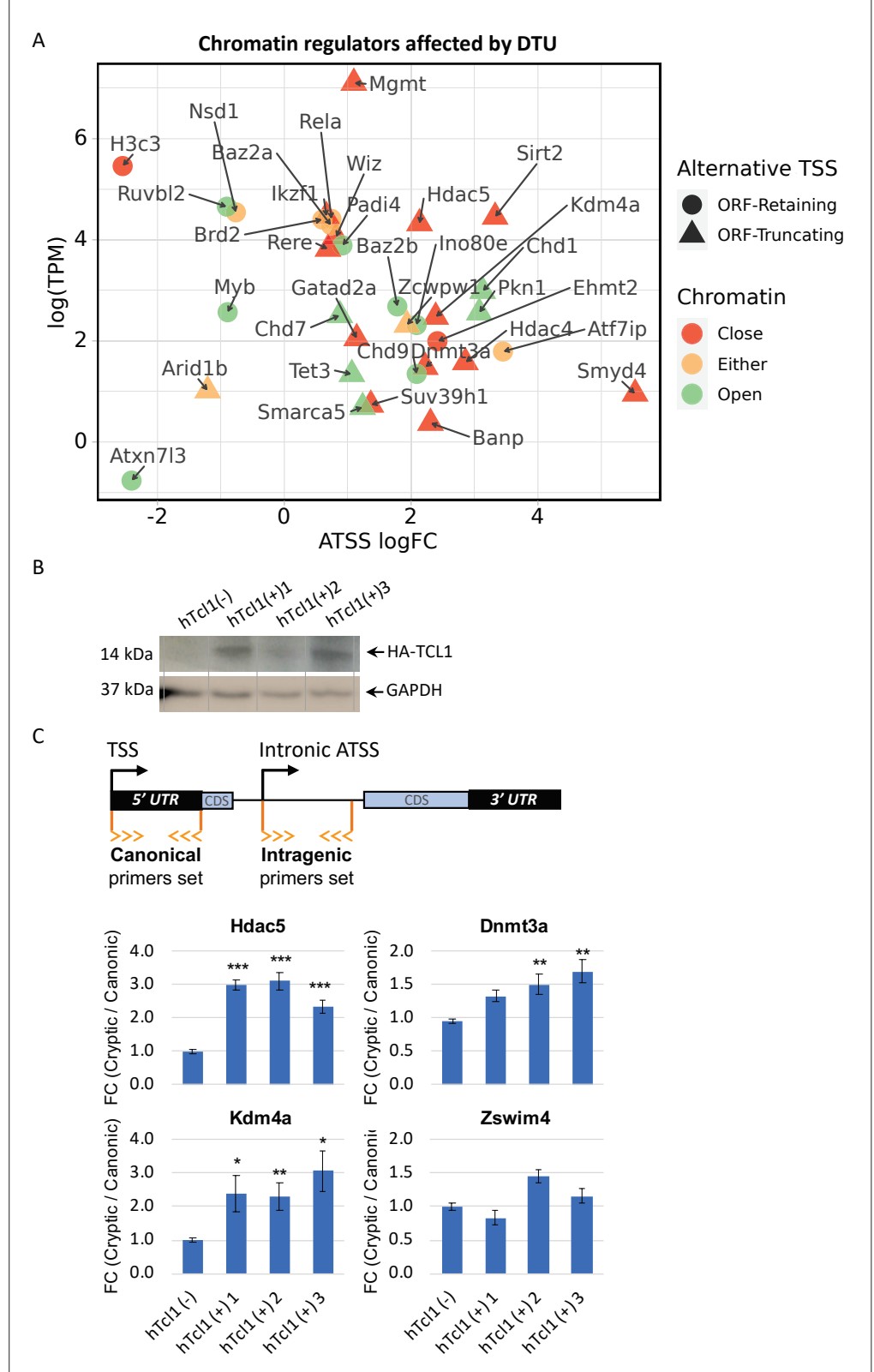

**Figure 3.** Cryptic promoters of closed chromatin epigenetic regulators and the role of hTCL1. (**A**) Dot plot of ORF-retaining (circle) and ORF-truncating (triangular) alternative transcription start sites (TSSs) within genes associated with close (red), open (green), or either (yellow) chromatin states. For evaluating the degree of promoter shifting (DTU), the data are presented by expression (log2 TPM) against log FC of alternative TSS (ATSS). Positive and

*Figure 3 continued on next page*

*Figure 3 continued*

negative ATSS log FCs refer to Eμ-Tcl1 up- and downregulated ATSS, respectively. (**B**) Stable mouse embryonic fibroblast (MEF) cell lines expressing exogenous hTCL1 were verified by western blot with anti-HA antibody and anti-GAPDH as loading control. (**C**) Exogenous expression of hTCL1 induces cryptic promoters. A scheme illustrating the canonical and cryptic promoter and the design of RT-qPCR primers to the 5′ UTR sequences originated by canonical TSS and intra-genic (intronic) alternative TSS. The results are presented as the ratio between cryptic and canonical levels. Statistically significant differences are denoted by asterisks as follows: *p < 0.05, **p < 0.01, and ***p < 0.001.

The online version of this article includes the following source data and figure supplement(s) for figure 3:

**Source data 1.** Western blot of MEF stable clones expressing HA-tagged human TCL1.

**Figure supplement 1.** 5′ RACE of alternative transcription start sites (TSSs) revealed by Cap Analysis of Gene Expression (CAGE).

Deregulation of the chromatin affects the accessibility of transcription factor (TFs) in promoter and enhancer regions. Since CAGE provides expression data of genes along with their promoter and enhancer regions, we investigated the contribution of TFs to the activation of cryptic promoters in Eμ-Tcl1. By generating position frequency matrices (PMSF) of TFBS motifs (*Fornes et al., 2020*, mouse core collection) we could appreciate the apparent differences in the occurrence of TFBS in ORF-retaining and -truncating alternative TSSs, differentially transcribed in Eμ-Tcl1 (*Figure 4A*). Shown by PCA2, TFBS frequencies mainly differ between up- and downregulated ORF-truncating alternative TSS, explaining 38.6% of the variation (*Figure 4A*). Using Fisher's exact test, we found 63 TFBS enriched in the promoters of the upregulated alternative TSSs compared to the promoters of the canonical upregulated TSS in Eμ-Tcl1 (*Figure 4B*). When we combine these data with the list of upregulated TFs in Eμ-Tcl1 (*Supplementary file 3*), we found that upregulated levels of the proto-oncogenes *Ets2* and *c-Myc* correspond to the highly enriched *Ets*-related and bHLH motifs, respectively (*Figure 4B*).

Activation of the *c-Myc* oncogene is the hallmark of many hematopoietic malignancies (*Ferrad et al., 2020*). As *c-Myc* is among the upregulated TFs in our CAGE data of Eμ-Tcl1 CLL and is predicted to regulate intra-genic APs, we asked whether *Tcl1* itself can activate *c-Myc*. Using the hTCL1 MEFs clones, we found that the levels *c-Myc* are dramatically upregulated in all the hTCL1 expressing clones, indicating that its activation in Eμ-Tcl1 B CLL is most likely intrinsic.

As many of the upregulated alternative TSSs in Eμ-Tcl1 are associated with active enhancers (*Figure 2—figure supplement 1B*), we analyzed TFBS enrichment also in the upregulated enhancers and found nine enriched motifs (*Figure 4D*). Remarkably, the TFs linked to these nine enriched motifs are all upregulated in the Eμ-Tcl1 (*Supplementary file 3*) and include *Mef2b*, *Pbx3*, *Maff*, *Nfil3*, *Atf3*, *Baft3*, *Tfap4*, and *Arid3a*. These findings strongly suggest that these enhancers and their

**Table 1.** Differential expressed chromatin-regulating genes.

| Symbol | log FC | Adj. p value | Protein |
| --- | --- | --- | --- |
| *Chd2* | −1.28 | 0.04 | Chromodomain helicase DNA-binding protein 2 |
| *Chd3* | −8.29 | 0.01 | Chromodomain helicase DNA-binding protein 3 |
| *Cbx4* | −1.69 | 0.02 | Chromobox 4, E3 SUMO ligase |
| *Kmt2c* | −1.2 | 0.03 | Lysine methyltransferase 2C |
| *Kmt5b* | −1.47 | 0.03 | Lysine methyltransferase 5B |
| *Hdac11* | −13.79 | 0.02 | Histone deacetylase 11 |
| *Hdac9* | −4.59 | 0.02 | Histone deacetylase 9 |
| *Kat6a* | −1.04 | 0.03 | Lysine acetyltransferase 6A |
| *Kdm7a* | −1.27 | 0.02 | Lysine demethylase 7A |
| *Smarca2* | −2.13 | 0.03 | SWI/SNF related |
| *Asf1b* | 4.36 | 0.032 | Anti-silencing function 1B histone chaperone |

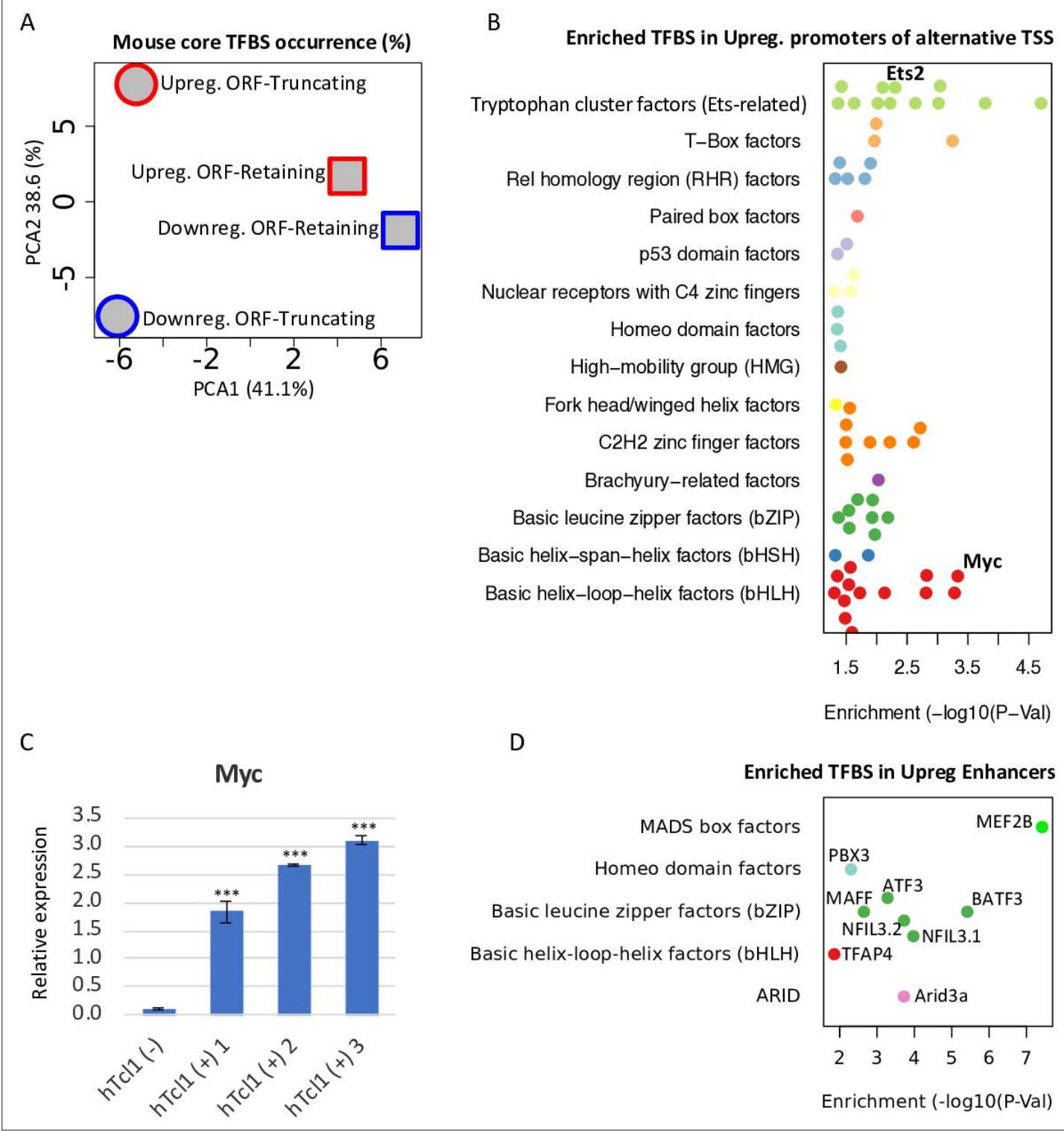

**Figure 4.** Upregulation of TFs targeting cryptic promoters and enhancers. (**A**) A principle component analysis (PCA) analysis of motif occurrence (%) of 119 transcription factor-binding sites (***Fornes et al., 2020***, mouse core collection) searched in 923 promoters (−1000 upstream and +100 downstream to transcription start site [TSS]) of upregulated (red) and downregulated (blue), ORF-retaining (squares) or ORF-truncating (circles) alternative TSSs. PCA1 and PCA2 explain 38.6% and 41.1% of the total variation, respectively. (**B**) TFBS enrichment analysis of activated alternative promoters (−1000 upstream and +100 downstream to TSS) relative to the overall activated promoters. Among the enriched TFs, *Ets2* and *c-Myc* levels are upregulated in Eµ-Tcl1 (***Supplementary file 3***) and are indicated. (**C**) Analysis of the *c-Myc* expression in the stable mouse embryonic fibroblast (MEF) clones expresing hTCL1 by RT-qPCR. Statistically significant differences are marked with asterisks ***p < 0.001. (**D**) TFBS enrichment analysis of enhancers associated with alternative promoters relative to overall induced enhancers. The levels of indicated enriched TFs are upregulated in Eµ-Tcl1 (see ***Supplementary file 3***).

corresponding upregulated transcription factors act as inducers of DTUs in Eµ-Tcl1. Altogether, these findings expand the oncogenic activities of hTCL1 to deregulation of diverse chromatin factors, activation of the *c-Myc* oncogene and enhancer-specific TFs, which cooperate to induce intra-genic cryptic promoters.

## Polysome-CAGE analysis of DTU in Eµ-Tcl1 CLL B cells reveals high translatability of intra-genic (cryptic) APs and impairment of epigenetic regulation

Considering that the extensive changes in TSSs are predicted to generate transcript isoforms that vary in their 5′ UTR and translation start site, we examined whether the Eµ-Tcl1-specific mRNAs originated from DTUs are translated into proteins. To this end, we performed polysome profiling of Eµ-Tcl1 splenic B cells and generated CAGE libraries from the fractions representing polysome-free (Free), light polysomes occupying two to four ribosomes (Light), and heavy polysomes (Heavy) occupying over four ribosomes per mRNA transcript (*Figure 5A*). The polysome profile of the Eµ-Tcl1 CLL cells shows a high monosome to polysome ratio (*Figure 5A*), reminiscent of a state of translation initiation halt, suggesting that protein synthesis capacity is limited in these cells. 3.39 million CTSSs were mapped and clustered to a consensus set of 30,246 promoters corresponding to 11,465 genes. Reproducibility analysis showed high coefficient scores calculated using the Pearson pairwise-correlation test (*Figure 5—figure supplement 1A*). Similar to the total RNA CAGE samples, most TSSs of the polysome profiling samples mapped with the highest expression level to the core promoter region as opposed to the highly frequent but poorly expressed intergenic and intronic TSSs (*Figure 5—figure supplement 1C*). As a support for the eRNA prediction shown to be abundant in the intergenic and the intronic regions (*Figure 1C*), most intergenic and intronic TSSs were found in the polysome-free fraction (*Figure 5—figure supplement 1B, C*), confirming that those entities are noncoding RNAs transcribed from active enhancers. To determine the TE of isoforms generated by alternative TSSs of multi-TSS genes, we calculated the ratio between the TSS counts in Heavy + Light fractions and the Free fraction. To explore the consequences of alternative TSS on TE, we defined 'Differential High-TE' only if its TE value was twofold higher than its paired TSS, and vice versa for 'Differential Low-TE' alternative TSS (illustrated in *Figure 5—figure supplement 1D*). Out of 13,924 per-gene TSS pairs, 2458 were assigned as 'Differential High-TE' and 2103 classified as 'Differential Low-TE'. After extracting 5′ UTR sequences, we tested our High/Low-TE calling by evaluating known translation regulatory features of 5′ UTR. We found that 'Differential Low-TE' TSSs tend to have longer 5′ UTR and to contain more uORF (*Figure 5B*), which are well-known inhibitory translation features.

Next, we examined the translatability of Eµ-Tcl1-specific transcript isoforms that arise from promoter shifts. By examining the CAGE data of total RNA and the polysome profiling fractions, we could determine the translatability of each isoform initiated by alternative TSS. Out of 543 DTUs found within the 5′ UTR, 142 were also differentially translated, of which 76 and 66 were transcriptionally up- and downregulated, respectively. Intriguingly, the vast majority of the transcriptionally upregulated DTU (60/76) are classified as High-TE, and most of the downregulated DTUs are classified as Low-TE (41/66) (*Figure 5C*), indicating for coordination of TSS selection with mRNA translation.

To assess the potential functional consequences of the intra-genic alternative TSSs, predicted to generate N-terminally truncated/modified protein isoforms, we calculated the ratio of CAGE tags between the intra-genic alternative TSS and the canonical TSS only from the polysomes fraction (translated transcripts). When the count of intra-genic alternative TSS in the polysome fraction exceed 50% of the canonical TSS, we consider this intra-genic alternative TSS to substantially affect the function of the canonical full-length protein. In total, we found 73 intra-genic alternative TSS that are upregulated in Eµ-Tcl1 and exceeding 50% of the canonical TSS and 168 intra-genic alternative TSS which their lower translatability will less likely to impact protein functionality (*Figure 5D*). GO analysis of these 73 genes is summarized in *Figure 5E* by plotting the level of promoter shifting (scored as inclusive log FC, bar height) and the translation level of an intra-genic alternative TSSs relative to the canonical TSSs, which provides insight into the potential vulnerability of their cellular activity. Genes have been clustered into the following cellular processes: 'apoptotic process', 'cell adhesion', 'chromatin regulation', 'DNA/RNA synthesis', 'generation of precursor metabolites and energy', 'ion channel', 'MAPK signaling', 'protein metabolism', 'signal transduction', and 'vesicle-mediated transport'. Interestingly, under 'chromatin regulation', we could find Eµ-Tcl1-specific truncated isoforms of chromatin

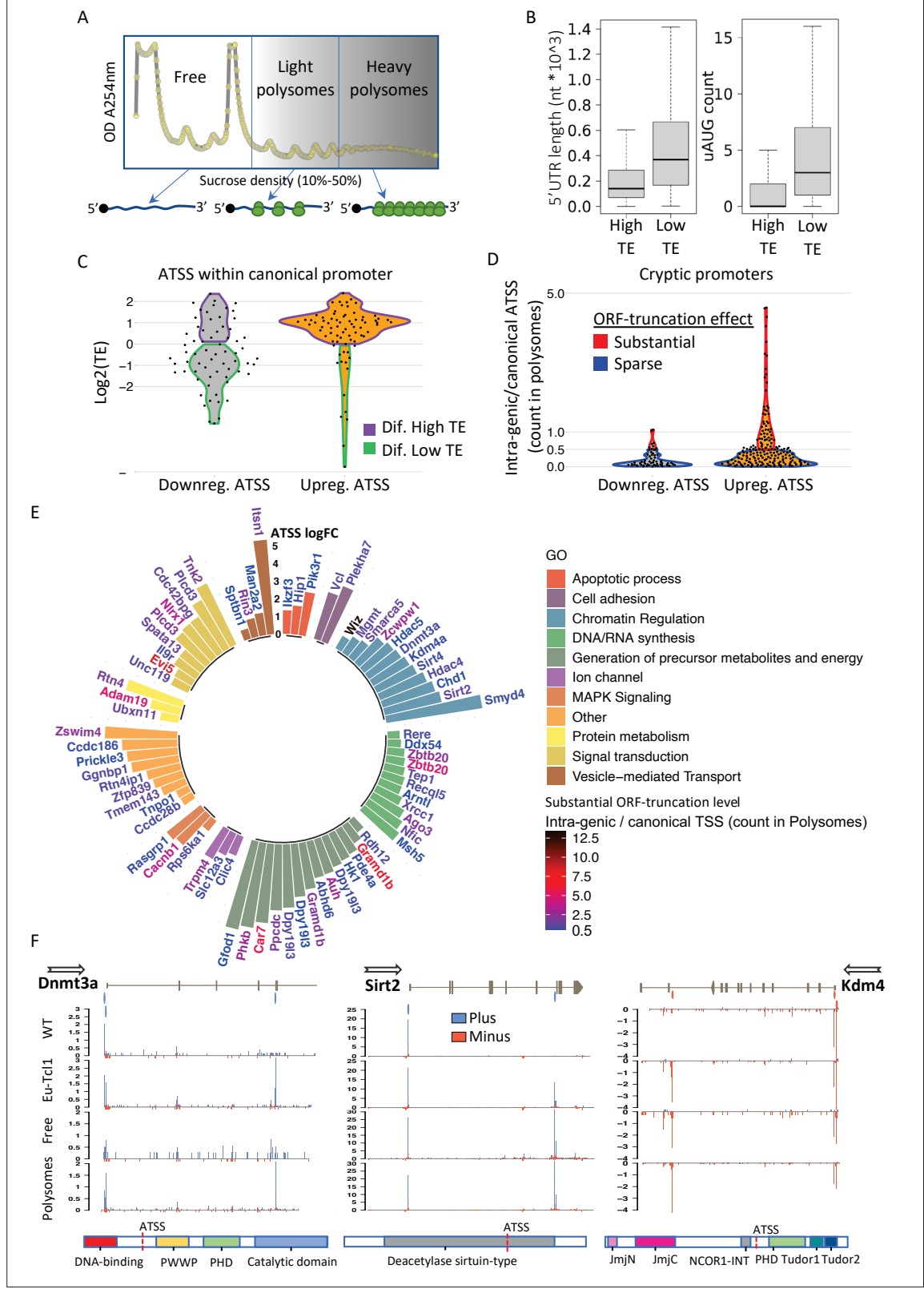

**Figure 5.** Translatability of ORF-retaining and -truncating alternative transcription start sites (TSSs). (**A**) Schematic presentation of polysome fractionation into three main fractions. Free fraction corresponds to nontranslated transcripts and Light and the Heavy fractions corresponds to intermediate and highly translated transcripts, respectively. (**B**) Boxplot of 5' UTR length and number of upstream AUG of transcript isoforms defined by high translation efficiency (High-TE) and low translation efficiency (Low-TE). (**C**) The TE distribution of differentially expressed ORF-retaining alternative

*Figure 5 continued*

TSSs (within canonical promoter) leading to differential High-TE (purple border) and differential Low-TE (green border) transcript isoforms. (**D**) ORF-truncating ATSSs that are either above (Substantial ORF truncation) or below (Sparse ORF truncation) 50% of the canonical TSS count in the polysomes fraction. (**E**) Gene ontology (GO) anlysis of genes with Eμ-Tcl1 upregulated and translated intra-genic ORF-truncating ATSSs. Bar heights present an inclusive log FC explaining the magnitude of promoter shifting, taking in consideration TSS FC between Eμ-Tcl1 and WT mice and FC between cannonical and intra-genic TSSs pairs, per gene. Gene names are colored by a relative scale of translation level between ORF-truncating ATSS and the canonical TSS, calculated by the ratio of intra-genic/canonical TSS counts in the polysome fractions. Blue colored gene refers to ORF-truncating ATSS that exceeds 50% (0.5) of the canonical representation in the polysomes fraction and dark red refers to the maximum of 12.5 FC higher representation in the polysomes fraction by the ORF-truncating ATSS. (**F**) Coverage peaks of TSSs of canonical and intra-genic cryptic promoters (indicated by blue and red ticks in the upper track) found in *Dnmt3a Sirt2* and *Kdm4a* genes. Blue and red coverage peaks represent transcription from the plus and minus strands, respectively. Eμ-Tcl1 and WT tracks presented TSS peaks of total RNA and Polysomes and Free tracks derived from Eμ-Tcl1 polysome profiling. On the bottom are schemes of each gene's protein domains in which the points of truncation due to intra-genic ATSS are marked.

The online version of this article includes the following figure supplement(s) for figure 5:

**Figure supplement 1.** Translatability of ORF-retaining and ORF-truncating alternative transcription start sites (TSSs).

remodelers and epigenetic players that might be involved in the aberrant TSS utilization found in Eμ-Tcl1 B cells. This class includes histone modifiers: *Smyd4*, *Sirt2*, *Sirt4*, *Hdac4*, *Hdac5*, *Kdm4a*, *Zcwpw1*, and *Wiz*; DNA methylation factors: *Dnmt3a* and *Mgmt* and chromatin remodeling factors: *Chd1*, *Smarca5*, and *Bcl7b*. In addition, under 'DNA/RNA synthesis' GO, we could find Eμ-Tcl1-specific truncated isoforms of DNA repair genes such as *Xrcc1*, *Recql5*, and *Tep1*, and direct transcription regulators essential for hematopoiesis that includes *Gfi1*, *Zbtb20*, *Ago3*, *Ssbp4*, *Ddx54*, and *Arntl* (*Figure 5E*). These cases of promoter shifting in Eμ-Tcl1 leads to the generation of highly translated protein isoforms with possible loss of function or dominate-negative activities. For example, Eμ-Tcl1-specific cryptic transcription (2.22 log FC) within the fourth intron of *Dnmt3a* leads to the loss of three exons (211 amino acids) that constitute its DNA-binding domain (*Figure 5F*), which is important to the DNA methylation activity (*Suetake et al., 2011*) and is generally associated with closed chromatin state (*Brocks et al., 2017*). The *Dnmt3a* cryptic TSS is enriched within Polysomes fractions, suggesting that *Dnmt3a* activity is substantially impaired (*Figure 5F*, left). A promoter shifting in Sirtuin 2 (*Sirt2*, 3.32 log FC), a type III histone deacetylase (*Hdac*), interrupts the deacetylase domain positioned between residues 65 and 340 that generates a truncated isoform with higher translatability compared to the canonical isoform which is enriched in the polysomes fraction (*Figure 5F*. middle). Together with HDACs, lysin demethylase (KDMs), erasers of histone three lysin-9 and lysin-36 methylation marks of active transcription are known to be corecruited to ensure gene silencing (*Mozzetta et al., 2015*). DTU found in lysine-specific demethylase 4a (*Kdm4a*) has an Eμ-Tcl1-specific cryptic transcription leading to a truncated isoform that is equally presented in polysome fractions as the canonical full-length isoform (*Figure 5F*, right). Thus, with five truncated members of the HDAC and KDM families (*Hdac4*, *Hdac5*, *Sirt2* and *Sirt4*, *Kdm4a*) and dramatic downregulation of *Hdac11* and *Hdac9* (−13.79 and −4.59 log FC, respectively, *Table 1*), it appears that Eμ-Tcl1 B cells are characterized by reduced eraser activities of histone acetylation and methylation, to maintain open chromatin state.

## Identification of CNY initiating trinucleotides as translation inhibitory motif associated with metabolic stress signature

The polysome-CAGE data enable the determination of the exact first nucleotide of every transcript and its impact on TE. We first determined the frequencies of TSS nucleotides in our Eμ-Tcl1 data and found that the frequency of A and G (47% and 36%, respectively) is higher than the C and T (9% and 4%, respectively), as expected (*Figure 6A*). Yet, the difference between purines to pyrimidines is substantially more significant. For example, in Eμ-Tcl1, purines account for 83%, but in MEFs, only 64% (*Tamarkin-Ben-Harush et al., 2017*), suggesting an increased preference of purines as start sites in B CLL cells. We next determined the TE of transcripts with different initiating nucleotides and observed that those initiated with a 'C' have a substantially lower TE compared to the A, G, and T (*Figure 6B*). To examine whether the lower TE associated with the cap-proximal C is linked to the well-known TOP element (CYYYY), we analyzed the TE of the first trinucleotides and found that almost all C initiating trinucleotides have a relatively lower TE, including inhibitory trinucleotides that are part of the TOP element such as CCT and CTT, but also others that deviate from the TOP such as CAC, CAT, and CTG. Interestingly, among the C-initiating triplets, those with a pyrimidine (CNY) in the third position are

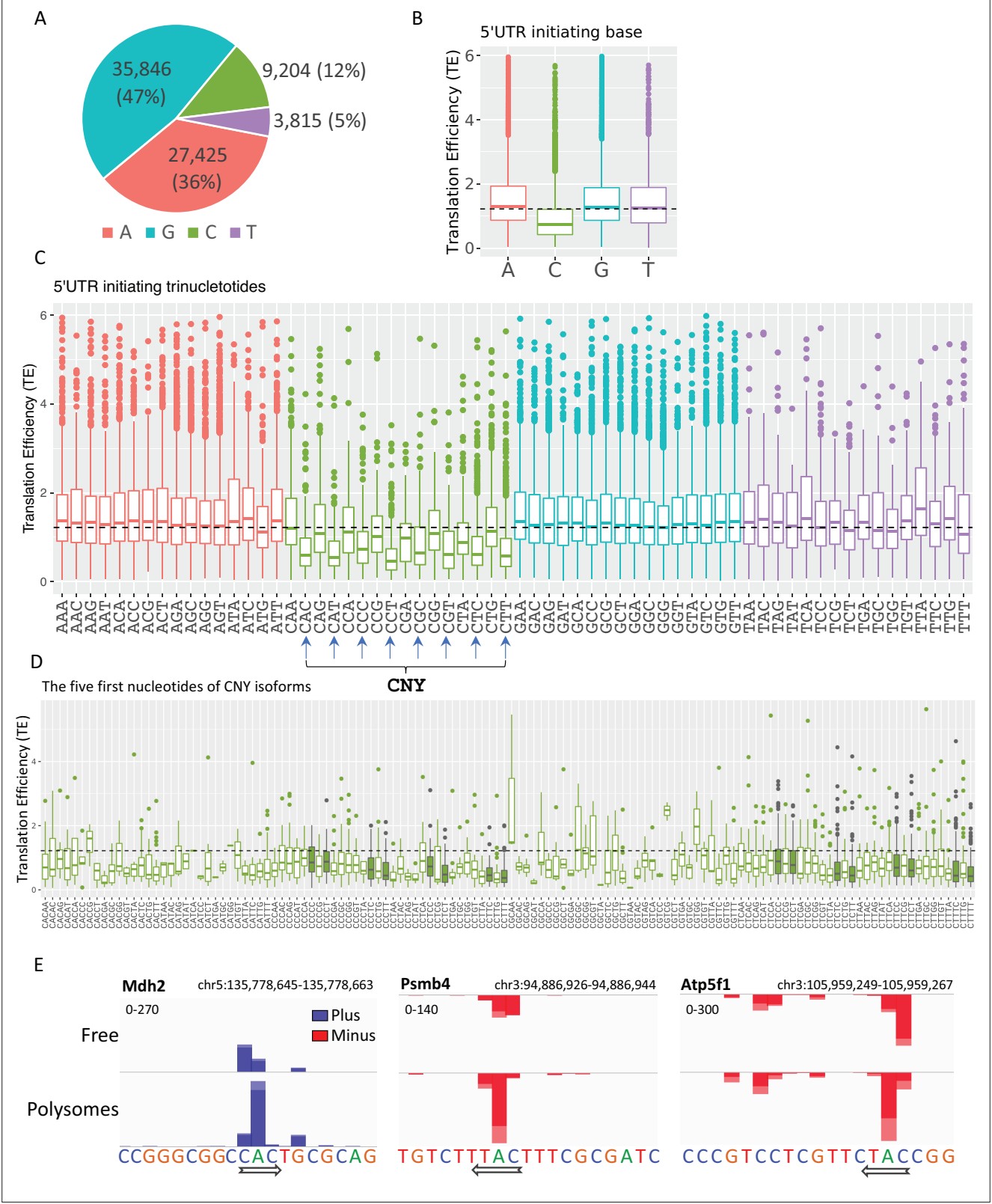

**Figure 6.** The impact of transcription start site (TSS) nucleotides on translation efficiency (TE). (**A**) The frequency of initiating TSS nucleotide in Eµ-Tcl1. (**B**) Boxplots of TE of isoforms that differ in their TSS first nucleotide. The horizontal dashed line marks the overall median value. (**C**) Boxplots of TE of transcripts that differ in their first three nucleotides. CNY trinucleotides are marked by arrows. (**D**) Boxplots of TE of the first five nucleotides of the CNY-initiating transcripts. The nucleotides corresponding to the TOP element are marked with filled boxes. All the data presented in this figure are the mean

*Figure 6 continued on next page*

*Figure 6 continued*

of the two independent replicates. The bottom and the top whiskers represent 5% and 95% of the distribution, respectively. (**E**) Examples representing the effect of the first nucleotides on TE in *Mdh2*, *Psmb4*, and *Atp5f1* genes. CAGE-derived TSS (CTSS) tags per million (TPM) is plotted separately for plus (blue) and minus (red) strands of the polysome-free and polysomal fractions with a scale set for the two tracks shown in the upper-left corner. Direction of transcription is marked by an arrow located at the first CTSS.

The online version of this article includes the following figure supplement(s) for figure 6:

**Figure supplement 1.** Boxplots of transcription efficiency (TE) of isoforms differ in their initiating transcription start site (TSS) single nt (**A**) and first three nucleotides (**B**) are distributed and colored by initiating Adenosine, Cytosine, Guanin and Tyrosine.

associated with a much lower TE (*Figure 6C*). Importantly, the nucleotides that follow the CNY do not seem to contribute to the TE suppression (*Figure 6D*), indicating that the CNY context is the predominant feature. *Figure 6E* presents three examples of TSSs adjacent to each other within the same promoter region that display differential Polysome to Free ratio. In the *Mdh2* gene, the reads of the two C in the CAC triplet are more enriched in the Free than the Polysomal fractions, while the opposite is seen for the reads that start with the A. The same trends are also seen in *Psmb4* and *Atp5f1*, of which the (A)s are more highly translated than the Cs. As the 5′ UTR generated from these TSSs is almost identical, these examples highlight the important contribution of the first nucleotides to TE.

A previous study that analyzed the relationship between the first nucleotide and the relative TE also reported that 'C' initiating transcripts are associated with lower TE, particularly following energy stress (*Tamarkin-Ben-Harush et al., 2017*). We reanalyzed these previous data and found that in energy-starved MEFs, the CNY is also associated with more significant inhibition of the TE (*Figure 6—figure supplement 1A, B*). Thus, the higher monosome-to-polysome ratio (*Figure 5A*) and the association of 'CNY' with reduced TE are consistent with Eμ-Tcl1 B CLL cells being under metabolic stress.

## CLL B cells are less sensitive to inhibition of epigenetic and translation regulators

To examine how CLL cells cope with the limited activities of epigenetic regulators, we isolated and cultured CLL and non-CLL B cells from Eμ-Tcl1 mice as well as B cells from healthy mice and treated them with inhibitors against the downregulated epigenetic factors HDAC3A, HDAC2A, KDM4A, and eIF4E in a dose–response assay. We found that the non-CLL or healthy B cells were inhibited by the HDAC2A and eIF4E inhibitors in a dose-dependent manner, whereas the CLL cells displayed reduced sensitivity (*Figure 7*). Interestingly, none of the B-cell types was affected by inhibitors against HDAC3A and KDM4A at the concentrations used. These findings suggest that the CLL cells are less dependent on the activities of these limiting factors.

## Discussion

In this study, we used the Eμ-Tcl1 mouse model of CLL to investigate the global changes in TSSs and AP usage and their impact on the protein content of cells undergoing oncogenic transformation. By comparing the TSSs landscape of normal and CLL B cells, our findings revealed a dramatic induction of APs in malignant CLL B cells with a particularly high prevalence of intra-genic promoters predicted to generate N-terminally truncated or N-terminally modified proteins. Importantly, a major class of intra-genic APs activated in the Eμ-Tcl1 CLL cells is the 'closed chromatin' epigenetic regulators gene subset, predicted to generate a loss of function or dominant-negative isoforms. Enhanced chromatin accessibility due to downregulation of epigenetic gatekeepers is well known to promote transcription initiation from intra-genic cryptic promoters sites (*Brocks et al., 2017*; *Wei et al., 2019*; *Mozzetta et al., 2015*; *Hennig and Fischer, 2013*; *Wei et al., 2020*; *Hennig et al., 2012*). Thus, intra-genic promoter shifts in B CLL follow a feed-forward loop, in which activation of N-terminally truncated closed chromatin modifiers further enhance the cryptic promoter activity. Additionally, the regulatory regions of the intra-genic-cryptic initiation sites (proximal promoters and the nearby enhancers) are targeted by the *c-Myc* oncogene and several transcription factors that are specifically upregulated in Eμ-Tcl1 B CLL. Thus, the combined loss of epigenetic regulators and the induced expression of a specific subset of transcription factors underlie the dramatic TSSs and AP usage changes in Eμ-Tcl1 B CLL. Notably, these dysregulated gene expression features are most likely intrinsic to the Eμ-Tcl1

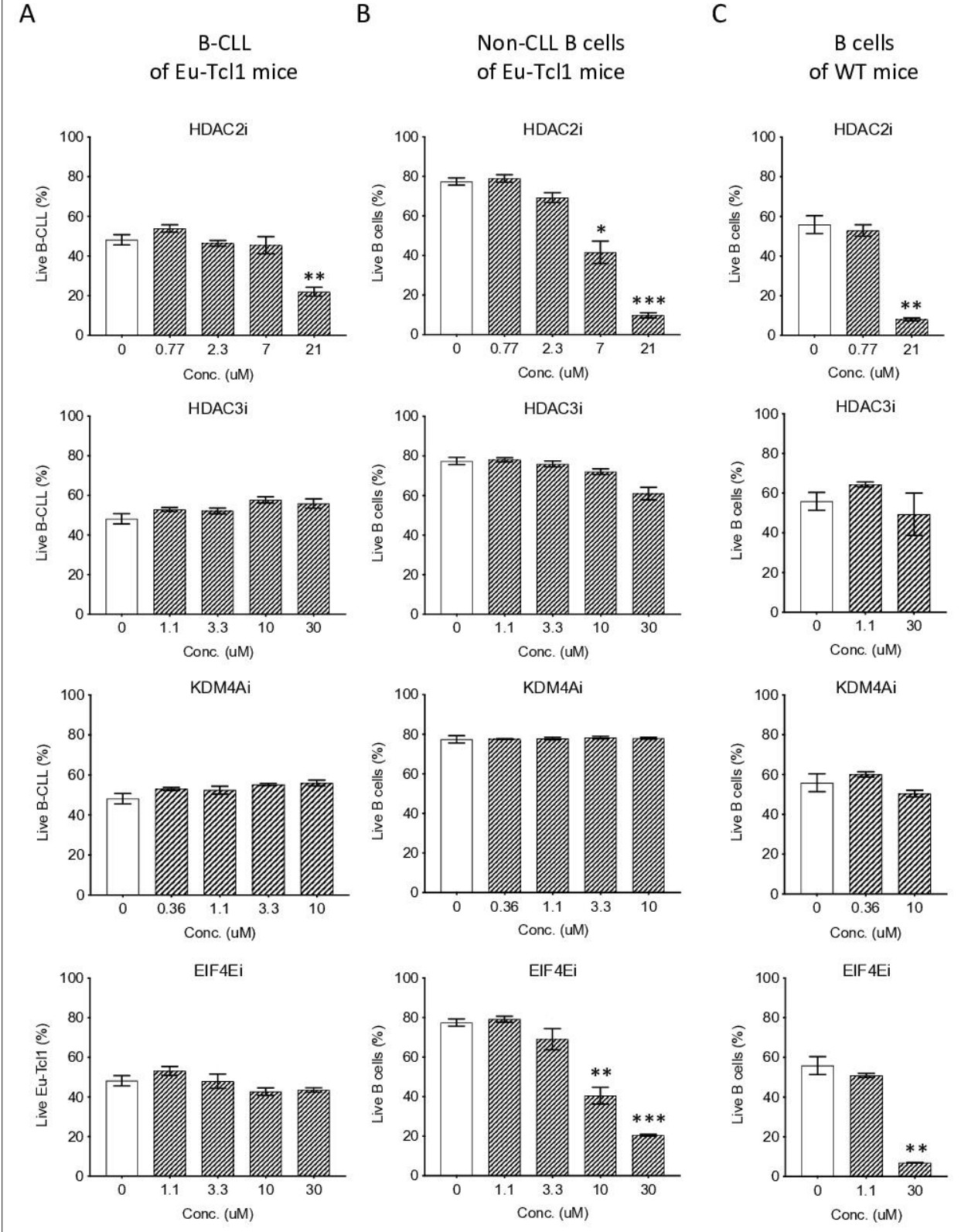

**Figure 7.** Eμ-Tcl1 B cells are relativlely resistant to inhibitors of affected pathway. Survival assay of B-CLL of Eμ-Tcl1 mice (**A**), non-CLL B cells of Eμ-Tcl1 mice (**B**), and B cell of WT mice (**C**) treated with inhibitors (i) targeting HDAC2, HDAC3, KDM4A and eIF4E. The significance between DMSO (vehicle control) and inhibitor-treated cells is indicated by asterisks (*p < 0.05, **p < 0.01, and ***p < 0.001).

B CLL, as exogenous expression hTCL1 in MEFs is sufficient to drive the induction of intra-genic promoter of epigenetic regulators and to upregulate *c-Myc*.

Our findings are consistent with a recent study that reported changes in promoter utilization in multiple cancer types (*Demircioğlu et al., 2019*). As this study used RNA-seq data to infer promoter activities, these data are substantially less accurate for estimating intra-genic promoter activity especially when it is embedded within internal exons or introns. Thus, our CAGE data of healthy and malignant B cells provided further insights on intra-genic promoter activities, their impact on the translatome, and the mechanism underlying their activation.

A previous study described decreased methylation levels in Eμ-Tcl1 mice and in CLL patients and reported a strong interaction between TCL1 and de novo DNA methyl transferases DNMT3A and 3B along with inhibition of the enzymatic activity (*Palamarchuk et al., 2012*), suggesting for direct inhibition of de novo methylation during leukemogenesis. In addition, TCL1 promotes the activity of the Alpha Serine/Threonine-Protein Kinase (AKT) signaling which also reduces DNMT3A activity (*Popkie et al., 2010*; *Yang et al., 2019*). Thus, activation of the AKT by the constitutively overexpressed TCL1 is likely to further reduce DNMT3A activity. Our findings complement these observations by uncovering another mechanism of impairment of DNMT3A activity via activation of an internal promoter and generation of large amounts of truncated inactive protein (*Figure 5F*, left). All these mechanisms are likely to inhibit DNMT3A and DNA methylation in these cells in an additive manner. Inhibition of DNMT3A together with HDACs is sufficient to deregulate chromatin and induce cryptic promoters within genomic regions that normally are silenced (*Brocks et al., 2017*; *Wei et al., 2019*; *Mozzetta et al., 2015*; *Hennig and Fischer, 2013*; *Wei et al., 2020*; *Hennig et al., 2012*). We therefore propose that downregulation of DNMT3A directly and indirectly by TCL1 is the initial trigger for the feed-forward loop that induces cryptic promoters in Eμ-Tcl1 CLL cells.

The use of polysome-CAGE enabled us to assess the potential contribution of the N-terminally truncated/modified protein isoforms derived from intra-genic promoters to the functionality of the intact protein. This analysis reveals substantial impairments in several regulatory and metabolic pathways, including chromatin regulation and energy metabolism, that are expected to contribute to the transformed phenotype. Remarkably, upon analysis of the relationship of the exact cap-proximal nucleotides and TE, we observed that all 'C' initiating transcripts display lower TE compared to the A, G, and T. Furthermore, among the 'C' initiated transcripts, those that start with CNY display even lower TE relative to the other Cs. The effect of the 'C' on the TE is nicely demonstrated in the examples in which adjacent nucleotides from the same promoter region display dramatic differences in polysome occupancy (*Figure 6E*). In these instances, the 5' UTR length and sequence are almost identical. Considering that translation downregulation of the well-known C-initiating TOP element is the hallmark of various metabolic stresses and mTOR inhibition (*Meyuhas and Kahan, 2015*), and the previous observation of downregulation of all C-initiating nucleotides following energy stress (*Tamarkin-Ben-Harush et al., 2017*), we infer that the Eμ-Tcl1 cells display a clear metabolic stress signature. Furthermore, the reanalysis of the first trinucleotide context of the energy-starved MEFs also identified CNY as the most inhibitory signal and the resemblance of the Eμ-Tcl1 CLL to metabolically stressed cells. Thus, our findings expand the regulatory sequences that mediate the effect of stress on translation beyond the TOP element. Such differences in the TE of initiating nucleotides were linked to reduced eIF4E levels upon energy stress, as its cap-binding activity is influenced by the identity of the first nucleotide (*Tamarkin-Ben-Harush et al., 2017*).

Considering the profound effect of TCL1 on multiple major regulatory pathways (*Figure 5*) and the tolerance of TCL1 expressing cells to the inactivation of these processes (*Figure 7*), it is reasonable that the most effective way to treat these malignancies is to target the TCL1 itself instead of the affected pathways. While TCL1 inhibitors have not yet been discovered, identifying such compounds should be an important therapeutic direction against TCL1-driven malignancies. Notably, the discovery of highly translated N-terminally modified proteins forms the basis for the identification of new, CLL-specific antigens that can be utilized for specific targeting of the CLL cells. Combining inhibitors against TCL1 and immunotherapy could be a promising strategy against CLL.

In summary, the ability of *Tcl1* to promote activation of intra-genic cryptic promoters of epigenetic regulators results in a feed-forward loop in which the loss of function of 'closed chromatin' inducers, by expression of N-terminally truncated isoforms, further enhance the activities of the cryptic promoter. This feature of *Tcl1* and its ability to boost *c-Myc* oncogene levels expand the current knowledge on

*Tcl1* transforming capacities and its multiple modes of regulation. The utilization of the polysome-CAGE not only verified the expression of N-terminally modified protein isoforms but also uncovered the restricted protein synthesis capability of Eμ-Tcl1 B CLL as a point of vulnerability that can be exploited as a drug target. Our findings form the basis for future work on approaches to interfere in tumorigenic processes mediated by TCL1 in TCL1-overexpressing leukemias.

## Materials and methods

### Mice

C57BL/6wt were purchased from Harlan Biotech Israel (Rehovot, Israel). Transgenic Eμ-TCL1 mice (*Bichi et al., 2002*) were kindly provided by CM Croce (The Ohio State University, Columbus, OH, USA). TCL1 mice were backcrossed for several generations to C57BL/6 mice to obtain TCL1 mice with the same WT background (co-isogenic). Mice were used when they reached progressed illness at the age of 1 year. All procedures were approved by the Animal Research Committee at the Weizmann Institute (IACUC approval number 06120721-3).

### Isolation of primary splenic B cells

Healthy and CLL splenic B cells were isolated from 1-year-old WT and Eμ-TCL1 mice by positive B-cell selection with CD19 magnetic beads (CD19 MicroBeads, Miltenyi Biotec, cat. 130-121-301). Briefly, spleens were squashed, treated with red blood lysis buffer (homemade) and filtered in phosphate-buffered saline (PBS) using 40-mm cell strainers. The cells were then counted and the B cells were purified according to the manufacturer's protocol. Splenocytes subjected for polysome profiling were isolated using buffers (above) supplied with 100 μg/ml Cycloheximide (Sigma-Aldrich) to form 80S–mRNA complex.

### Flow cytometry

Analysis of the surface expression of CD5 and CD19 on primary mature WT and Eμ-Tcl1 B cells was monitored by flow cytometry. Briefly, $1 \times 10^6$ cells were washed with PBS supplemented with 0.5% bovine serum albumin and 0.1% sodium azide and stained with FITC-conjugated anti-CD5 (eBioscience, cat. 11-0051-85) and with PE-Cy7-conjugated anti-CD19 (Invitrogen, cat. 25-0193-82) antibodies. Stained cells were washed and analyzed by flow cytometry (FACSCanto II, BD Biosciences), and the quantification of the measurements was analyzed using FlowJosoftware (TreeStar). The purity of WT mature B cells was estimated to be between 95% and 99%, and positive gated Eμ-Tcl1 B cells were estimated between 80% and 85%.

### Polysome profiling

Eμ-Tcl1 B cells were washed with cold buffer containing 20 mM Tris pH 8, 140 mM KCl, 5 mM MgCl$_2$, and 100 μg/ml cycloheximide. The cells were collected and lysed with 500 μl of the same buffer that also contains 0.5% Triton, 0.5% DOC, 1.5 mM DTT, 150 units RNAse inhibitor (Eurx, cat. E4210), and 5 μl of protease inhibitor (Sigma-Aldrich, cat. 78430). The lyzed samples were vortexed and centrifuged at 12,000 × *g* at 4°C for 5 min. The cleared lysates were loaded onto 10–50% sucrose gradient and centrifuged at 38,000 rpm in a SW41 rotor for 105 min at 4°C. Gradients were fractionated, and the optical density at 254 nm was continuously recorded using ISCO absorbance detector UA-6. The collected samples were then pooled to create three main fractions: Polysome-free (Free), Light (2–5 ribosomes), and Heavy (over five ribosomes).

### RNA extraction

Total RNA from WT and Eμ-Tcl1 purified B cells and the RNA from the pooled polysomal fractions (described above) were extracted in two biological replicates using TRIzol reagent (Invitrogen, cat. 15596026) following the manufacture's instructions. After phase separation, the aqueous phase was diluted back in TRizol reagent (1:3) to proceed with RNA purification by zymodirect RNA kit (Zymo research, cat. R2051). The quality of RNA samples was assessed using Agilent 2200 TapeStation (Agilent Technologies, USA) to evaluate RNA integrity (RIN), and purity was measured by Qubit 4 fluorometer (ThermoFisher, cat. Q32852).

## CAGE library preparation and sequencing

Five µg RNA samples were subjected to library preparation of Cap Analysis Gene Expression CAGE using CAGE protocol adapted for Illumina sequencing (*Takahashi et al., 2012*). Specifically, to promote complementary DNA (cDNA) synthesis through GC-rich sequences in the 5′ untranslated regions (5′ UTR), the reaction was carried with the presence of D-trehalose (Sigma-Aldrich cat. T0167) and D-sorbitol (Wako, cat. 19803755) at high temperature. Capped RNAs of RNA–DNA hybrids were Biotinylated (Vector lab, cat. SP1100) and purified by MPG streptavidin beads (Takara, cat. 6124A) followed by RNAse (Promega, cat. M4261) digestion to release cDNAs corresponding to the 5′ ends of the original mRNA. 5′ linkers harboring barcodes sequences and EcoP15I (NEB, cat. R0646S) recognition site were ligated, and second-strand synthesis was performed. Then, fragments of the first 27 bp of the 5′ UTR were produced by EcoP15l digestion and ligated to 3′ linkers containing the Illumina primer sequence. The resultant CAGE tags were amplified by polymerase chain reaction (PCR), purified and sequenced by Hiseq 2500 (Illumina) with the addition of 30% PhiX spike-in to balance the low complexity of the 5′ ends of the CAGE libraries.

## CAGE data preprocessing and mapping

Preprocessing pipeline of CAGE raw reads included quality control check using FSTQC, removal of Phix reads (Escherichia virus PhiX174, NC_001422.1) and rRNA CAGE tags using Bowtie2 (2.3.5.1), demultiplexing and linkers trimming using FASTX-toolKit (0.0.13). Clean CAGE tags were aligned to the mouse reference genome (mm10) using Bowtie2 with the `--very-sensitive` preset (set as -D 20R 3N 0L 20 -i S,1,0.50), and the resulting SAM files were converted to sorted BAM files using SAMTools (1.9). BAM files of Light and Heavy polysome fractions were merged to form one BAM file representing all translated fractions, marked as Polysomes. BAM files were then loaded into the CAGEr package (v1.32 in the Bioconductor environment, v3.12) and uniquely mapped tags were used for calling CTSSs setting parameters as follows: sequencingQualityThreshold = 20 and mappingQualityThreshold = 20. As part of CTSS mapping, CAGEr performed corrections for the CAGE-specific G-bias caused by template-free guanine incorporation upstream to the true TSS by the reverse transcriptase used for cDNA synthesis (*Kröber et al., 2002*). Metagene analysis for profiling CTSS coverage was performed using the 'metagene' package in the Bioconductor environment. Eµ-Tcl1 and WT CTSS locations were compared to the reference dataset for TSSs (refTSS) of the mouse (v3.1) published by RIKEN institute (*Abugessaisa et al., 2019*).

## CAGE tag clustering, quantification, and analysis

Most of the CTSS analysis was performed using CAGEfightR package (1.10.0) 4, which uses several R packages from the Bioconductor project, v3.12. BIGWIG files that were exported by the CAGEr were loaded into CAGEfightR and closely spaced CAGE tags were clustered into unidirectional and bidirectional tag clusters (TCs) using the slice-reduce approach to define the global TSSs and enhancer candidates, respectively. After counting TCs with less than 1 TPM in the two Eµ-Tcl1 or WT CAGE libraries were removed in further analysis to remove very lowly expressed TC and to obtain likely biological relevance. Active enhancers predicated as balanced bidirectional transcription of capped eRNAs with a threshold of Bhattacharyya coefficient set to 0.95, where value of 1 correspond to perfectly balanced sites. TSS–enhancer physical interactions predicted by distance (50 kbp) and positive correlation (Pearson >0, p < 0.05) as previously shown by *Andersson et al., 2014*. TxIDs, with their structure models (e.g., promoter, UTR, CDS, exon, intron, and antisense) and GeneID were assigned to each TC using the two genome-wide annotation packages, 'TxDb.Mmusculus.UCSC.mm10.knownGene' (*Team and Maintainer, 2019*) and 'org.Mm.eg.db' (*Carlson, 2019*), of the Bioconductor project, respectively. TC widths measured between 10% and 90% of the IQR of pooled CAGE tags to remove CAGE tags that greatly extend the TC width without contributing much to the total expression. TC widths were classified by the bimodally distributed sharp (1–10 bases) and broad (11–100 bases) TCs to be separately analyzed further. TC- and gene-level DE analyses were performed by Deseq2 package with a significance factor (alpha) cutoff set to 0.05 and with default setting of limma packages of the Bioconductor project, respectively. DTU analysis was performed using the edgeR diffSpliceDGE method on the gene subset holding more than TSSs corresponding to more than 10% of total gene expression.

## Motif-based sequence analysis for TF-binding sites

In order to know what transcription factors might be involved in the regulation of alternative TSSs, we downloaded DNA-binding motifs as position frequency matrices (PFMs) from the core collection of JASPAR database (*Fornes et al., 2020*) and matched it against promoter regions defined as −1000 bp and +100 to the peaks of CAGE TCs of TSSs and enhancers. We used TFBSTools (*Tan and Lenhard, 2016*) and motifmatcher (*Schep, 2022*) packages of the Bioconductor project, to obtain and find matches of motifs PFMs in promoter sequences, respectively. Using Fisher's exact test, we have been able to see if a motif significantly co-occurs in upregulated alternative TSS and enhancers.

## 5′ RACE

For the 5′RACE we used the template switching RT enzyme mix (NEB), which make use of the Template Switching Oligo (TSO) with known sequence followed by three RNA Guanidines. Using gene-specific RT primers (listed in *Supplementary file 4*), the reverse transcriptase reaches to the 5′ ends of the RNA template and switches to the TSO template by extending few nontemplated nucleotides on the cDNA 5′ end which then anneal to the rGrGrG sequence of the TSO to generate cDNA of 5′ leader sequences of all targeted genes. Based on 5′ RACE cDNA templates, we designed qPCR primers specifically targeting the 5′ UTRs of the truncated isoforms of all targeted genes.

## Eµ-Tcl1 adoptive transfer model

Generation of this mouse model was performed as previously described (*Hofbauer et al., 2011*). Briefly, Eµ-Tcl1 mice approximately 12 months of age, with a malignant cell population higher than 60% in the PB were sacrificed. Their spleens were excised, and $4 \times 10^7$ cells resuspended in PBS−/− were injected into the tail vein of 6- to 8-week-old WT recipient mice. Progression of the disease was followed in the PB by using flow cytometry for the IgM/CD5 population. Mice with >30% IgM+/CD5+ cells were considered to be diseased and were used for further analysis.

## Inhibitor assay

The survival of CLL B cells and healthy B cells was tested under the inhibitory effect of drugs targeting the activity of HDAC type II (TMP269, Cayman cat. 17738), HDAC type III (Thiomyristoyl, Cayman cat. 19398), KDM4A (NSC636819, Sigma-Aldrich, cat. 531996), and DNMT3A (Decitabine, MCE cat. HY-A0004). B cells isolated from Eµ-Tcl1 adoptive transfer and WT mice as described above. Triplicates of 2 million cells (per well) distributed in a 24-well tissue-culture plate treated with a serial four incremented concentrations of the above inhibitors (in DMSO). Following 48 hr, cells were washed with cold PBS, and resuspended in Annexin V Binding Buffer (BioLegend) at a concentration of 0.25–1.0 × $10^7$ cells/ml. FITC Annexin V (BDPharmingen cat. 556419) was added to the samples with anti-CD19 and anti-CD5 for 20 min in the dark on ice. Cells were washed twice with Annexin V Binding Buffer (centrifuge 1400 rpm for 5 min, discard the supernatant), then resuspended in 300 µl of Annexin V Binding Buffer with 7-AAD (BDparmingen cat. 88,981E) Viability Staining Solution. Cells immediately were analyzed by flow cytometry.

## Generation of MEFs expressing human *Tcl1*

*Tcl1* cDNA was cloned in pCRUZ-HA expression plasmid (SantaCruz) using the restriction-free cloning method (*Unger et al., 2010*) and cotransfected into MEFs from a WT mouse (a gift from Benois Viollet, INSERM, Paris) (*Tamarkin-Ben-Harush et al., 2017*). After G418 selection for 3 weeks, positive colonies were analyzed were propagated and then analyzed by western blot using anti-HA antibody.

## Acknowledgements

Funding: This work was supported by grants from the Israel Science Foundation (#843/17); Israel Cancer Association (#20220034); the Minerva Foundation (#713877) and by Weizmann Institute internal grants from Estate of Albert Engleman; Estate of David Levinson. RD is the incumbent of the Ruth and Leonard Simon Chair of Cancer Research.

## Additional information

### Funding

| Funder | Grant reference number | Author |
|---|---|---|
| Israel Science Foundation | 843/17 | Rivka Dikstein |
| Israel Cancer Association | 20220034 | Rivka Dikstein |
| Minerva Foundation | 713877 | Rivka Dikstein |
| Weizmann Institute of Science | Estate of Albert Engleman | Rivka Dikstein |
| Weizmann Institute of Science | Estate of David Levinson | Rivka Dikstein |
| Israel Science Foundation | 1642/20 | Idit Shachar |

The funders had no role in study design, data collection, and interpretation, or the decision to submit the work for publication.

### Author contributions

Ariel Ogran, Data curation, Software, Formal analysis, Supervision, Validation, Investigation, Methodology, Writing - original draft; Tal Havkin-Solomon, Shirly Becker-Herman, Keren David, Investigation, Methodology; Idit Shachar, Formal analysis, Funding acquisition, Investigation, Methodology, Writing - review and editing; Rivka Dikstein, Conceptualization, Formal analysis, Supervision, Funding acquisition, Investigation, Writing - original draft, Project administration

### Author ORCIDs

Ariel Ogran  http://orcid.org/0000-0002-9411-3537
Rivka Dikstein  http://orcid.org/0000-0002-6251-4723

### Ethics

This study was performed in strict accordance with the recommendations in the Guide for the Care and Use of Laboratory Animals of the Weizmann Institute of Science. All of the animals were handled according to approved Institutional Animal Care and Use Committee (IACUC) protocols of the Weizmann Institute of Science. The protocol was approved by the Committee on the Ethics of Animal Experiments of the Weizmann Institute of Science. All surgery was performed under sodium pentobarbital anesthesia, and every effort was made to minimize suffering.

### Decision letter and Author response

Decision letter https://doi.org/10.7554/eLife.77714.sa1
Author response https://doi.org/10.7554/eLife.77714.sa2

## Additional files

### Supplementary files

• Supplementary file 1. Gene ontology (GO) pathways affected by differential transcription start site (TSS) usage. GO enrichment analysis by the GeneAnalytics tool of GeneCardSuite, performed over a total of 923 gene set affected by differential TSS usage, generating alternative isoforms either up- or downregulated in Eu-Tcl1 mice. SuperPath matching score is based on the binomial distribution.

• Supplementary file 2. Mouse genome informatics (MGI) phenotypes affected by differential TSS usage. Enrichment analysis of MGI phenotypes performed by the GeneAnalytics tool of GeneCardSuite over a total of 923 gene set affected by differential transcription start site (TSS) usage, generating alternative isoforms, either up- or downregulated in Eu-Tcl1 mice. MGI phenotypes matching scores are based on the binomial distribution.

• Supplementary file 3. Transcription factors upregulated in Eu-Tcl1 A list of gene-level upregulated (positive log(FC)) transcription factors in Eu-Tcl1 mice.

• Supplementary file 4. A list of oligos/primers used in 5' RACE and qPCR methods.

• Transparent reporting form

## Data availability

Sequencing data have been deposited in GEO under accession code GSE194265.

The following dataset was generated:

| Author(s) | Year | Dataset title | Dataset URL | Database and Identifier |
|-----------|------|---------------|-------------|-------------------------|
| Ogran A | 2022 | Polysome-CAGE of TCL1-driven chronic lymphocytic leukemia revealed multiple N-terminally altered epigenetic regulators and a translation stress signature | https://www.ncbi.nlm.nih.gov/geo/query/acc.cgi?acc=GSE194265 | NCBI Gene Expression Omnibus, GSE194265 |

The following previously published dataset was used:

| Author(s) | Year | Dataset title | Dataset URL | Database and Identifier |
|-----------|------|---------------|-------------|-------------------------|
| Ulitsky I, Dikstein R | 2017 | Cap-proximal nucleotides via differential eIF4E binding and alternative promoter usage mediate translational response to energy stress | https://www.ncbi.nlm.nih.gov/search/all/?term=GSE93981 | NCBI Gene Expression Omnibus, GSE93981 |

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
