## [Editor Report]

Results presented in this article suggest that T-Cell Leukemia/Lymphoma 1 (TCL1) protein promotes alternative transcription site selection in chronic lymphoid leukemia, resulting in the production of N-terminally truncated versions of chromatin regulators and induction of expression of transcription factors including c-MYC. Finally, the authors provide evidence that TCL1 drives translational reprogramming akin to that observed under metabolic stress. Notwithstanding technical limitations that obscured direct detection of N-terminally truncated protein variants, it was found that this study is of broad potential interest inasmuch as it provides previously unappreciated evidence that TCL1 orchestrates epigenetic, transcriptional, and translational rewiring in leukemic cells. Based on this, the study by Ogran et al. should be of significant interest across a number of biomedical research disciplines ranging from regulation of gene expression to cancer research.

---

## [Decision Letter]

**Decision letter after peer review:**

Thank you for submitting your article "Polysome-CAGE of TCL1-driven chronic lymphocytic leukemia revealed multiple N-terminally altered epigenetic regulators and a translation stress signature" for consideration by *eLife*. Your article has been reviewed by 3 peer reviewers, including Ivan Topisirovic as the Reviewing Editors and Reviewer #1, and the evaluation has been overseen by Kevin Struhl as the Senior Editor.

Essential revisions:

1) It was found that the evidence of expression of putative N-terminally truncated proteins is required to fully support the proposed model. Notwithstanding that the overarching characterization of N-terminally truncated proteins across the proteome may be out of the scope of the paper, it was thought that the authors should at least provide evidence of in-depth-discussed proteins (e.g., chromatin modifiers). Moreover, it was thought that orthogonal techniques (e.g., 5'RACE) should be used to confirm the results of CAGE studies.

2) Link between TCL1-dependent alterations in transcription site selection and/or promoter usage and chromatin remodelling was found to be largely correlative. Based on this, it was thought that the authors should provide more mechanistic data to support the role of apparent TCL1-dependent alterations in chromatin factors and perturbations in chromatin structure.

3) As outlined in the individual reviews, some important issues were observed pertinent to bioinformatic analyses. It was also thought that several important controls were missing. For instance, a more detailed rationale should be provided for cut-offs/thresholds used in bioinformatic approaches. The reproducibility of the results between the two CAGE replicates was also not clear. An apparent lack of application of unique molecular identifiers in the CAGE approach was also observed. In addition, concerns were raised regarding the application of log-ratios to calculate translational efficiency. Finally, there was a lack of loading controls in some of the Western blots.

4) In general, it was thought that the studies using epigenetic and translational inhibitors (targets of which were thought to be in some cases at least partially misattributed) do not show that the observed effects are TCL1-specific. Based on this, it was deemed that the authors should either provide additional data supporting the tenet that TCL1 allows adaptation to insults caused by pharmacological targeting of the epigenetic modifiers and/or translational apparatus or significantly temper claims in this section of their study.

*Reviewer #1 (Recommendations for the authors):*

- It was found that it would be crucial to test the expression of truncated proteins experimentally. It is understandable that it would be unreasonable to request this on a global scale, but these experiments should be carried out on a selected set of proteins that are mentioned in the study (e.g., chromatin remodelers, PD-L2, Chd1, fbxo5) to further support the author's model.

- Although the polysome-CAGE approach to establish engagement of different 5'UTR isoforms on polysomes was highly appreciated, it was thought that the stoichiometry of ORF-truncated/retained vs. corresponding full-length proteins should be established for at least the subset of affected genes.

- The cut-off of 50% for polysome engagement of a given 5'UTR isoform on corresponding protein function seems arbitrary. To this end, it was thought that the functional characterization of the proteins whose length may be affected by alternative promoter selections/transcription start site selection would be advantageous. The latter should provide some insights into the relevance of the alternative promoter usage/TSS selection to TCL1-driven phenotypes. This was thought to be important as is it remains unclear what is the contribution of proposed mechanisms to TCL1-driven phenotypes, whereby the prior evidence shows that TCL1 plays a rather pleiotropic function in CLL whereby it engages a number of oncogenic pathways. It is understandable that the full phenotypic characterization of the proposed model is likely to be out of the scope of this study, but the functional characterization of at least a few of the selected proteins appears to be warranted.

- In line with the comment above, the link between TCL1-dependent ORF-retention and/or truncation of chromatin remodelers and alterations in chromatin appears to be purely correlative. The authors should either provide mechanistic evidence for the causative relationship between these phenomena (e.g., by focusing on Dnmt3a) or tone down their claims pertinent for this part of the study.

- There is no evidence that the observed effects of the inhibitors (figure 7) are TCL1-specific (e.g., would the same effects be observed in TCL1-overexpressing MEFs?). These data are rather preliminary and insufficient to support the tenet regarding the adaptation of CLL B cells to epigenetic and translational reprogramming. Literature showing the potential effectiveness of targeting suggested pathways/factors in CLL (e.g., eIF4E) was also largely ignored. Moreover, the author referred to MNK1/2 inhibitor as eIF4E inhibitor, which was found to be inappropriate. This part of the study is significantly weaker than the rest, and the authors should consider either expanding it or substantially toning down their claims.

- Some issues were noted pertinent to data analysis. For instance, the calculation of translational efficiency using log ratios that were previously demonstrated to be confounded by potential spurious correlations (PMID: 21115840). Addressing these issues along with providing more details related to the analyses that were performed throughout the study seems to be required.

*Reviewer #2 (Recommendations for the authors):*

1. Page 3 last paragraph starting with "The transcription initiation.…..opposite biological functions." Would benefit from referring to the literature for the different claims.

2. On page 4, the feed-forward loop is introduced. This would benefit from being more clearly written.

3. I likely missed this, but it was not clear to me which TSS was selected as the alternative. Was it the one close to the annotated transcription start site?

4. On page 6 bottom paragraph "TSSs were enriched at the core promote…": what does the enrichment refer to? The length of these sequences? What were the odds-rations and the statistics for these enrichments?

5. Figure 2G. It is claimed that truncating ATSS are more commonly upregulated in Tcl1 overexpressing cells. Can this be shown more clearly? Can one calculate statistics for this?

6. On page 8 "TFBS frequencies mainly differ between….". But this is in component 2 which is not the main component.

7. Figure 6E requires stats.

8. The Results section "CAGE tag clustering, quantification and analysis" includes a few sentences lacking words.

9. Is there an adjustment for the correlation p-values when performing TSS-enhancer analysis (described on page 18). It seems that false discovery rates rather than unadjusted p-values should be used.

*Reviewer #3 (Recommendations for the authors):*

Comments below are ordered following the flow of the manuscript:

1) The authors perform total CAGE-Seq and polysome CAGE-Seq in duplicates. Please, show the correlation between duplicates.

2) Figure 2D-F: Please, validate these alternative TSS usage examples by semi-quantitative PCR or Northern blot of wt versus Eµ-Tcl1 cells.

3) Figure 3B: Please, provide the loading control for this Western blot, and an assessment of the level of over-expression compared to endogenous Tcl1.

4) Figure 3B-C: To really assess whether changes in alternative TSS usage are due to Tcl1 over-expression, and not to environmental signals associated with cancer progression, the authors should perform CAGE-Seq in the MEF cell lines over-expressing Tcl1 that they have built and compare the overlap of detected TSS events with those found in B-cells. Right now, only 3 cases have been confirmed and a general conclusion cannot be reached.

5) Figure 3C: Please, indicate the nº of independent biological experiments and replicates performed.

6) Figures 4C-D: Please, provide supporting Western blots.

7) Figure 5: The authors propose that Tcl1-mediated changes in TSS usage in chromatin modulators contribute to chromatin openness in Eµ-Tcl1 B cells. Can the authors directly test this by assessing global levels of histone acetylation or by performing ATAC-Seq of WT vs Eµ-Tcl1 B cells? Can the authors check the presence of stable, N-terminally truncated versions of some of the chromatin remodelers in Figure 5F by Western blot?

8) Sup Figure 5C and differential TE explanation in the main text:

The formula in Sup Figure 5C explains the concept of TE, but not of differential TE mentioned in the main text where a reference TSS (paired TSS) is claimed. From the drawing, it seems that differential TE is counted twice for any given two promoters, and this is confusing. What happens with genes containing more than 2 promoters? Which promoters do you compare with to decide on high or low TE? A reference promoter per gene should be chosen (e.g., the canonical promoter) and the data normalized to that reference.

9) Figure 6E: Here the authors compare TSSs that start at almost the same location within a given gene, as they are only different by one nucleotide ('TSSs adjacent to each other'). How do the authors know that this one nucleotide differences are real, and not due to methodological CAGE-Seq errors? Why would such one nucleotide differences have profound effects on polysome association? Could this be an artifact of library preparation or bioinformatic treatment of CAGE-Seq? The conclusion that Eµ-Tcl1 cells are under metabolic stress just because of the similarities regarding 1st nucleotide choice between this study and that of ref 21 seems speculative.

10) Figure 7: What can one conclude from the fact that inhibitors against 2 out of 3 of the epigenetic factors downregulated in Eµ-Tcl1 B cells have anyway no effect on the viability of B-cells no matter their origin? I'm not sure this figure tells much. Furthermore, controls that the drugs are really working (e.g., by measuring that they work on other cell lines) would be necessary.

---

## [Author Response]

Essential revisions:1) It was found that the evidence of expression of putative N-terminally truncated proteins is required to fully support the proposed model. Notwithstanding that the overarching characterization of N-terminally truncated proteins across the proteome may be out of the scope of the paper, it was thought that the authors should at least provide evidence of in-depth-discussed proteins (e.g., chromatin modifiers). Moreover, it was thought that orthogonal techniques (e.g., 5'RACE) should be used to confirm the results of CAGE studies.

As suggested, we now performed 5’ RACE to validate the CAGE results by picking four chromatin modifiers and examining the upregulation of the alternative TSS in Eu-TCl1 mice. The 5’ RACE was done using template switching RT enzyme mix (NEB), Template Switching Oligo (TSO) and a gene-specific reverse primer common to the canonical full-length isoform and the N’ terminally truncated one. In order to compare the transcription level of the truncated isoform between Eu-Tcl1 and WT mice, isoform-specific primers were used in the qPCR assay. The results of the 5’RACE-qPCR authenticate our CAGE-seq results, showing upregulation of the isoforms originating by alternative TSS of the four selected DNMT3a, CHD1, KDM4A and SIRT2 (see Figure 3—figure supplement 1 in the revised manuscript).

Notably, we also confirmed that these isoforms are efficiently translated to proteins by combining CAGE with polysome profiling (Figure 5). Other proteomic approaches such as mass spectrometry (MS) and western blot for detecting the N-terminally truncated/modified proteins are less suitable. With the MS, the peptides derived from N-terminally truncated proteins are shared with the full-length protein. In western blot, there are several limitations that include the finding of high-quality antibodies directed against the C-terminus of the candidate proteins, the ability to distinguish between truncated and proteolytically cleaved protein, and the ability to predict the new translation initiation site accurately, the ORF and the size of the protein isoform. This is particularly challenging since, in most cases, the new promoters are located within an intron. Nevertheless, we performed multiple western blots by purchasing antibodies against the C-terminus of selected proteins, but we could not identify the truncated protein with high certainty.

2) Link between TCL1-dependent alterations in transcription site selection and/or promoter usage and chromatin remodelling was found to be largely correlative. Based on this, it was thought that the authors should provide more mechanistic data to support the role of apparent TCL1-dependent alterations in chromatin factors and perturbations in chromatin structure.

By combining our data with data from the literature, a mechanism by which TCL^-^1 promotes the activation of cryptic promoters is emerging. We briefly referred to this mechanism in the discussion of the original manuscript. In the revised manuscript, we further expand it with more specific details. Specifically, a previous study reported decreased DNA methylation levels in Eµ-Tcl1 mice and CLL patients and demonstrated an interaction between TCL1 and the de novo DNA methyl transferases DNMT3A and 3B along with inhibition of their enzymatic activity (1), suggesting for direct inhibition of de novo methylation by TCL1 during leukemogenesis. In addition, it was reported that TCL1 promotes the Α Serine/Threonine-Protein Kinase (AKT) activity. As AKT signaling itself reduces Dnmt3a activity (2, 3), over-sensitization of the AKT by the constitutively overexpressed TCL1 is likely to reduce DNMT3A activity further. Our findings complement these observations by uncovering another mechanism of impairment of DNMT3A activity via activation of an internal promoter and generation of large amounts of truncated inactive protein (Figure 5F, left). All these mechanisms inhibit DNMT3A and DNA methylation in these cells in an additive manner. We, therefore, propose that downregulation of DNMT3A directly and indirectly by TCL1 is the initial trigger for the activation of cryptic promoters in Eµ-Tcl1 CLL cells, which is further augmented by the activation of cryptic promoters of other chromatin regulators in a feed-forward loop and by activation of c-myc (see more details in the discussion).

(1) Palamarchuk, A., Yan, P.S., Zanesi, N., Wang, L., Rodrigues, B., Murphy, M., Balatti, V., Bottoni, A., Nazaryan, N., Alder, H. et al. (2012) Tcl1 protein functions as an inhibitor of de novo DNA methylation in B-cell chronic lymphocytic leukemia (CLL). Proceedings of the National Academy of Sciences of the United States of America, 109, 2555-2560.

(2) Yang, Qi, Wei Jiang, and Peng Hou. "Emerging role of PI3K/AKT in tumor-related epigenetic regulation." *Seminars in Cancer biology*. Vol. 59. Academic Press, 2019.

(3) Popkie, Anthony P., et al. "Phosphatidylinositol 3-kinase (PI3K) signaling via glycogen synthase kinase-3 (Gsk-3) regulates DNA methylation of imprinted loci." *Journal of Biological Chemistry* 285.53 (2010): 41337-41347.

3) As outlined in the individual reviews, some important issues were observed pertinent to bioinformatic analyses. It was also thought that several important controls were missing. For instance, a more detailed rationale should be provided for cut-offs/thresholds used in bioinformatic approaches.

As suggested, we now provide explicit reasonings for the bioinformatics cut-off/thresholds described in the method paragraph titled “CAGE tag clustering, quantification and analysis<milestone-start />״<milestone-end />. After counting CAGE reads into defined tag clusters (TC), we filtered out TCs with less than one tag per million (TPM) to remove very lowly expressed TC and obtain likely biological relevance. Where eRNA prediction is based on finding two balanced bidirectional (sense and antisense) TCs, we set the threshold of the Bhattacharyya coefficient (BC) to 0.95, where the value of 1 corresponds to perfectly balanced sites. In addition, when we measured the width of TCs of pooled CAGE tags, we used a 10-90% interquartile range (IQR) threshold to dampen the effect of possible straggler tags that can greatly extend the width of a TSS candidate without contributing much to expression. Deseq2-based results of differentially expressed TCs were independently filtered by setting the α argument to 0.05, which is the significance cutoff used for optimizing the independent filtering. For the analysis of differential TSS Usage (DTU) that was done by the diffSpliceDGE method of the edgeR package, we used a subset of genes with more than one TSS to analyze. By considering only TSS that correspond to more than 10% of total gene expression, we guarantee meaningful detection of promoter shifting events during CLL transformation.

The reproducibility of the results between the two CAGE replicates was also not clear.

We have added coefficient scores of CAGE library replicates derived from the RNA samples of Eu-TCL1 and WT, as well as CAGE library replicates from the polysome profiles of Eu-TCL1. (See Figure 1 —figure supplement 1A and Figure 5 – —figure supplement 1A). All replicates have shown high coefficient scores by the Pearson pairwise-correlation test, indicating the high reproducibility of our CAGE results.

An apparent lack of application of unique molecular identifiers in the CAGE approach was also observed.

Indeed, using the unique molecular identifiers (UMI) sequences for removing PCR-originated read duplicates became a gold standard and in 2017, the nanoCAGE protocol introduced the UMIs to the CAGE methodology. However, this method is claimed to be inferior to the classic CAGE method for the following reasons: (a) the template switching method used by nanoCAGE was shown to be sequence-dependent and, therefore, is potentially biased (Tang et al. 2013, *Nucleic Acids Res*
**41**: e44 10.1093/nar/gks1128) (b) the nanoCAGE was optimized for the use of a small amount of starting RNA (50 ng) that lead to low-complexity libraries with high levels of duplicates (Cvetesic, Nevena, et al.2018). Also, it was claimed by Cvetesic, Nevena, et al., that the synthesis of truly random UMIs is problematic and subject to variability, thereby obscuring its use. Taking all these into consideration, the sequencing errors of UMI sequences (Smith et al. 2017), our ability to start with a higher amount of 5 ug total RNA to reduce PCR cycles to a minimum of eight, and by that avoid amplification biases, we decided to follow the classic CAGE method.

In addition, concerns were raised regarding the application of log-ratios to calculate translational efficiency.

We wish to clarify that we did not use the log-ratios method for TE calculation as reported in (PMID: 21115840) and noted by Reviewer 1. Briefly, the calculation method we used is based on the ratio of the counts of the polysome fractions to the free fraction (see p. 9)

Finally, there was a lack of loading controls in some of the Western blots.

We have now added the missing loading control using antibodies against GAPDH (see Figure 3B).